# Flow-induced order–order transitions in amyloid fibril liquid crystalline tactoids

Hamed Almohammadi [1], Massimo Bagnani [1] & Raffaele Mezzenga [1,2✉]

Liquid crystalline droplets, also known as tactoids, forming by nucleation and growth within the phase diagram region where isotropic and nematic phases coexist, challenge our understanding of liquid crystals under confinement due to anisotropic surface boundaries at vanishingly small interfacial tension, resulting in complex, non-spherical shapes. Little is known about their dynamical properties, since they are mostly studied under quiescent, quasi-equilibrium conditions. Here we show that different classes of amyloid based nematic and cholesteric tactoids undergo order–order transitions by flow-induced deformations of their shape. Tactoids align under extensional flow, undergoing extreme deformation into highly elongated prolate shapes, with the cholesteric pitch decreasing as an inverse power-law of the tactoids aspect ratio. Free energy functional theory and experimental measurements are combined to rationalize the critical elongation above which the director-field configuration of tactoids transforms from bipolar and uniaxial cholesteric to homogenous and to debate on the thermodynamic nature of these transitions.

[1] Department of Health Sciences and Technology, ETH Zurich, Zurich, Switzerland. [2] Department of Materials, ETH Zurich, Zurich, Switzerland.
✉email: raffaele.mezzenga@hest.ethz.ch

Liquid crystalline droplets, or tactoids, form through self-organization of mesogens in concentrated aqueous suspensions[1,2] and are characterized by vanishingly low interfacial tension; the interfacial tension ($\gamma$) of tactoids based on filamentous colloids is of the order of $10^{-7}\,\mathrm{N\,m^{-1}}$, that is ~100,000 times smaller than typical water–oil emulsion[3]. While microfluidic has been widely used to study water–oil emulsions[4–7] and to produce monodisperse micron-sized droplets of liquid crystals in another immiscible liquid with interfacial tension in the order of water–oil emulsion[7], the hydrodynamics of liquid crystalline droplets remains poorly studied since most of the studies in these systems aimed at understanding primarily their thermodynamically equilibrium features[8]. The ability to manipulate the shape of liquid crystalline droplets by hydrodynamic forces using microfluidics allows studying their deformation-induced phase transitions[9] and tune tactoids self-assembly structure[10], and therefore provides a promising platform to design new materials[8]. Additionally, elucidating the hydrodynamic properties of liquid crystalline tactoids can broaden our understanding of soft active matter systems exploiting anisotropic directional interactions[11,12].

Tactoids form through nucleation and growth of anisotropic domains when the mesogens are dispersed at a concentration where isotropic (I) and nematic (N) phases coexist[1–3] and show various director field configurations depending on the intrinsic characteristics of the building blocks[1,3]. Thus, these systems are effectively a fascinating example of water-in-water emulsions with anisotropic elastic features and vanishing small interfacial tension, which endow tactoids a very unique and peculiar set of physical properties. To date, liquid crystalline tactoids have been found in dispersions of many biological rod-like systems including tobacco mosaic viruses[13], fd viruses[14], f-actin[15], carbon nanotubes[16], cellulose[17], and more recently amyloid fibrils[1,3,18]. Amyloid fibrils, generated from common food proteins, are an appealing system due to their high potential in designing new functional materials[19,20], but also due to their very rich liquid crystalline behavior[3,18]. As recently discovered[3,18], amyloid fibrils

show transition between six different equilibrium symmetries of the nematic field configuration: homogeneous, bipolar, radial nematic, uniaxial cholesteric, and radial cholesteric, with additional parabolic focal conics in bulk[18]. Such configurations result from the subtle interplay between anisotropic interfacial tension and anisotropic elastic forces in the droplet and, at equilibrium, the transitions between these different classes of tactoids are accurately predicted by either scaling or variational theories[3,18]. It is also important to mention the recent findings on the effects of the length and polydispersity of mesogens on the cholesteric pitch, showing that the pitch decreases with increasing amyloid fibrils length[1] while length polydispersity of mesogens enhances the twist elastic modulus[21]. However, far less is known regarding the behavior of the liquid crystalline droplets under a flow field, where—as we show below—the shape of the droplets can be significantly altered by flow field with order–order transitions from one symmetry to another. Studying liquid crystalline tactoids under hydrodynamic forces opens a window on a virtually unexplored physics where anisotropic tensorial elasticity coupled with the minimal interfacial tension is anticipated to bring to light unexpected effects and to deepen the understanding of emulsions characterized by anisotropic or very low interfacial tension, such as the elusive water-in-water emulsions[22].

By using amyloid fibril tactoids as a model system, here we first map flow-induced order–order transitions associated with a change in the director field configuration of the droplets during deformation. We then combine simple fluid droplet deformation theories with scaling concepts on the free energy functional theory, to rationalize our experimental findings on tactoids deformation and order–order transitions, providing a general framework to study and understand the behavior of liquid crystalline droplets under deformation.

## Results

**Microfluidic system to expose tactoids to extensional flow.** In order to deform different classes of liquid crystalline tactoids under extensional flow, we designed a microfluidic chip with

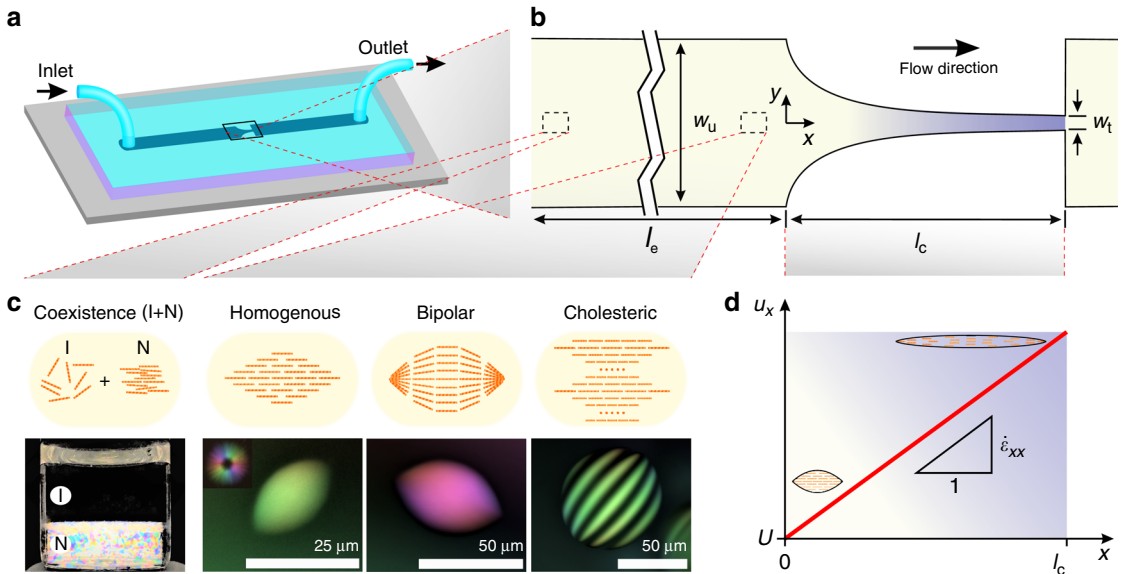

**Fig. 1 Microfluidic chip to expose different classes of tactoids to extensional flow. a** Schematic of the microfluidic chip. **b** The hyperbolic contraction geometry. The origin of the coordinate system is located at the contraction inlet with $x$ axis along the channel centerline. **c** A suspension of amyloid fibrils at concertation in the isotropic (I) + nematic (N) coexistence region is injected to the microfluidic chip. LC (liquid crystal)-PolScope images of the three classes of homogenous, bipolar, and cholesteric tactoids studied. **d** Schematic plot of the average velocity of flow, that varies linearly along the contraction zone to allow a constant extension rate ($\dot{\varepsilon}_{xx}$). The term $U$ denotes the flow speed in the straight channel before and after contraction zone.

hyperbolic contraction zone (Fig. 1a, b). An aqueous suspension of amyloid fibrils is prepared at a concertation set within the isotropic–nematic coexistence region, allowing the formation of various classes of tactoids. The mean length of the fibrils is $L_{f,m} = 303$ nm with $44 \leq L_f \leq 1500$ nm and standard deviation in normal distribution $\pm 194$ nm and lognormal distribution fitting parameters $\mu_{fitting} = 5.5 \pm 0$ and $\sigma_{fitting} = 0.6 \pm 0$; the mean height of the fibrils, which corresponds to the fibrils average diameter considering a cylindrical approximation, is $D_{f,m} = 2.5$ nm, with $1.0 \leq D_f \leq 6.2$ nm and standard deviation in normal distribution $\pm 0.7$ nm and lognormal distribution fitting parameters $\mu_{fitting} = 0.9 \pm 0$ and $\sigma_{fitting} = 0.3 \pm 0$ (Supplementary Note 1). The suspension is injected into the chip where the flow speed (controlled by syringe pump) and the length of the channel from the inlet to the contraction zone ($l_e$) are controlled to allow the solution, with the formation of different classes of tactoids, to equilibrate. The three main classes of tactoids that are studied here can be categorized by an increase in volume (calculated as $V \approx r^2 R$ by measuring major, $R$, and minor, $r$, axes of tactoids) and a decrease in aspect ratio ($\alpha = R/r$) as: homogenous nematic, bipolar nematic, and uniaxial cholesteric (Fig. 1c). The tactoids are classified based on their director field configurations[1,3]. In homogenous tactoids, the director field is aligned to the long axis of the tactoid. In the case of bipolar tactoids, the director field always follows the interface. The uniaxial cholesteric configuration is characterized by its typical striped texture and quantified by pitch value ($P$), which is two times the band-to-band distance. The configurations of the tactoids at rest are directly linked to their confined structures and the transitions between different classes of tactoids at equilibrium are theoretically predicted by Nystrom et al.[3] as a function of $V$ and $\alpha$; the pitch is found to correlate with the shape of the cholesteric droplet where upon an increase in $V/\alpha$, the pitch decreases asymptotically[3].

The tactoids, when passing through the hyperbolic geometry, experience extensional flow[23]. The width of the contraction zone, $w(x)$, with length ($l_c$), height or thickness ($h$), upstream width ($w_u$), and throat width ($w_t$), is shaped according to the function proposed in ref. [23] as $w(x) = x_1/(x_2 + x)$, where $x_1 = l_c w_u w_t/(w_u - w_t)$ and $x_2 = l_c w_t/(w_u - w_t)$. Accordingly, for a given volumetric flow rate ($Q$) and with the assumption that the shear flow induced by the bounding wall is negligible, the flow speed, $u_x = Q/[w(x)h]$, increases linearly as $x$ increases, leading to a constant extension rate, $\dot{\varepsilon}_{xx} = \frac{\partial u_x}{\partial x} = Q/(x_1 h)$, along the hyperbolic contraction zone ($0 \leq x \leq l_c$), Fig. 1d. To ensure that the assumption of negligible shear flow effect by the channel wall holds valid in the analysis, only the tactoids passing close to the centerline of the channel ($y = 0$) were analyzed.

Various dimensions of contraction geometry (see "Methods") and flow rates ($0.3 \leq Q \leq 1.8$ mm$^3$ h$^{-1}$) are used to induce different extensional rate $\dot{\varepsilon}_{xx}$, i.e., $0.004 \leq \dot{\varepsilon}_{xx} \leq 0.020$ s$^{-1}$. The flow speed in the straight channel before and after the contraction zone is $U = Q/(w_u h) = Q/(w_d h)$, where $w_d$ is the width of downstream channel that is equal to $w_u$ for all used geometries, and was always found in the range of $0.49 \leq U \leq 1.92$ μm s$^{-1}$ in the experiments. The microfluidic channel was fabricated with two different heights (or thicknesses) $h$, equal to 100 μm and 200 μm; see "Methods" for detailed information.

**Fate of liquid crystalline tactoids under flow field**. When injected in the microfluidic chip, the tactoids align their long axes parallel to the flow direction while approaching the contraction zone ($x < 0$). In the contraction zone ($0 < x < l_c$), the tactoids are subjected to extensional flow. As shown in Fig. 2, a multitude of complex events can be resolved by this approach. The most common event is the continuous stretch of homogenous tactoids

while maintaining their director symmetry, independently of the extension; in contrast, the bipolar and cholesteric tactoids undergo order–order transition to homogenous tactoids (Fig. 2a–c) at a critical extension. To distinguish different classes of tactoids, we examined the tactoids between crossed polarizers and tracked the texture of the tactoids during the elongation process. In the Supplementary Note 2, we further show how displaying different textures of the tactoids at different angles, relative to crossed polarizers, allows resolving unambiguously the deformation-induced order–order transitions of bipolar and cholesteric tactoids into homogenous tactoids. Right after the contraction ($l_c < x \lesssim 2l_c$), the tactoids rotate by ~90°, aligning their long axes perpendicular to flow direction, recovering their original starting phases. Such a parallel and perpendicular alignment of tactoids to the flow direction before and after the contraction zone, respectively, was shown previously for anisotropic cylindrical and disk-like colloidal particles[24]; however, we show here that similar behavior can be observed also in tactoids, extending the validity of the argument to larger length scales (Supplementary Note 3). Additional events include break-up and coalescence of the tactoids. Example of the break-up of a bipolar tactoid into a homogenous and a bipolar tactoids is shown in Fig. 2d, reminiscent of the liquid filament break-up as a function of its aspect ratio and rheological properties[25,26]. Coalescence of tactoids into larger volume can also occur when two tactoids approach simultaneously the contraction zone, as shown in Fig. 2e. Such a break-up and coalescence processes of tactoids demonstrate that extensional fluid flow can be used to manipulate tactoids volume by disentangling it from the mass transport between continuous and dispersed phase, which typically occurs during nucleation and growth. This is of critical importance in designing the systems with desired classes of tactoids. Note that the coalescence events shown here occur due to hydrodynamic focusing region of the selected geometry where the tactoids come together following the flow streamlines. The detailed discussion on coalescence and the formation of uniform bulk cholesteric phase (Fig. 2i) is beyond the scope of this paper and will be addressed in a separate detailed investigation.

A distinctive tract of the undergoing deformation mechanisms is the very low interfacial tension characterizing the tactoids. To highlight this further, control experiments with simple fluids (oil droplet in water–glycerol mixture) with an interfacial tension in the order of ~0.01 N m$^{-1}$ were performed and compared with those on the tactoids (Fig. 2f). To make a meaningful comparison, the relevant parameters in droplet deformation[27], i.e., droplet size, the viscosity of the droplet, viscosity of the medium, and accordingly the viscosity ratio of droplet to medium, as well as extension rate, were kept virtually identical (Supplementary Notes 4 and 5). Note that, although the viscosities of the liquid crystalline phases vary depending on the shear rate (Supplementary Note 4), for the sake of comparison and following the common assumption in the context of the droplet deformation[27], zero-shear viscosity values of liquid crystalline phases were considered here. The extension rate and droplet size are set to be equal to those of the corresponding cholesteric tactoid shown in Fig. 2c. When comparing Fig. 2c, f, it becomes clear that, while the tactoid reaches a length ratio of ~10 upon deformation, the oil droplet remains almost undeformed, highlighting the remarkable effects of very low interfacial tension of the tactoids.

**Deformation of tactoids under extensional flow**. The measurements of the deformations of tactoids under extensional flow are shown in Fig. 3a–c. In order to quantify the deviation of the shape of tactoid from its initial shape we use the Taylor deformation parameter, defined as[28]: $\mathcal{D} = (R - r)/(R + r)$, having zero

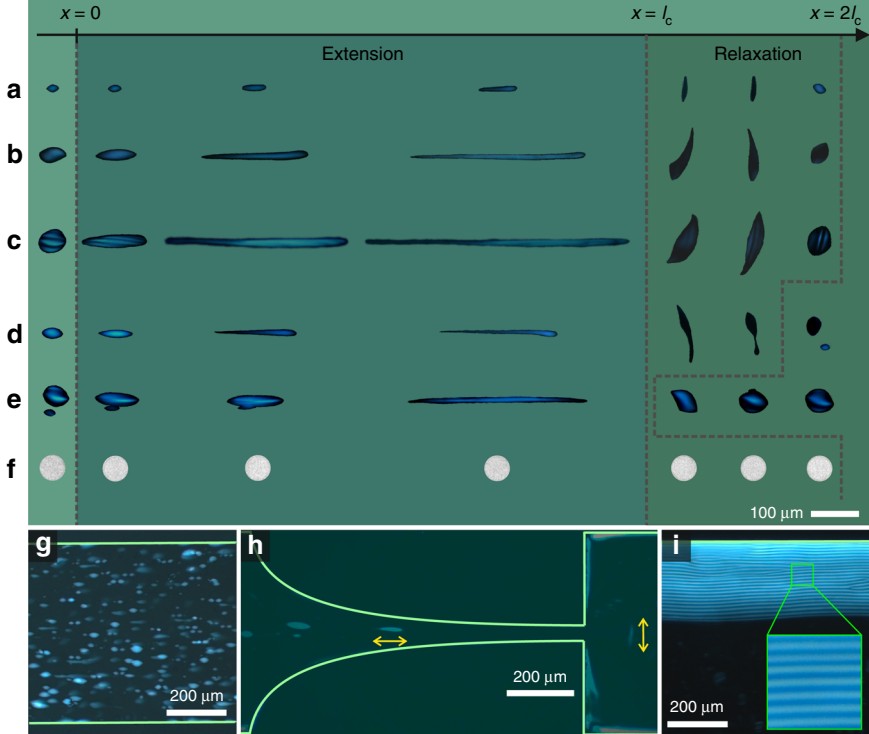

**Fig. 2 Fate of amyloid fibril tactoids under extensional flow field. a–c** The tactoids pass through the contraction zone (flow direction is from left to right) while having their long axes aligned with the flow direction. In the contraction zone ($0 < x < l_c$), the homogenous tactoids maintain their original phase (**a**), while the bipolar (**b**) and cholesteric (**c**) tactoids undergo order–order transitions to homogenous tactoids. After the contraction ($l_c < x \lesssim 2l_c$), the tactoids rotate ~90° and recover their original phase. **d** Break-up of a bipolar tactoid. **e** Coalescence of two tactoids resulting into a tactoid with larger volume. **f** A control experiment on oil-in-water droplets with interfacial tension in the order of ~0.01 N m$^{-1}$, i.e., orders of magnitudes higher than the liquid crystalline tactoids, illustrates how the oil droplet remains essentially undeformed through the same flow field conditions used to deform the tactoids. **g** Alignment of the tactoids in flow direction. **h** Rotation by ~90° of the tactoids exiting the channel. **i** Formation of a uniform bulk cholesteric phase as large as 200 μm with few disclination lines on the channel wall close to the exit of the channel, appeared by flowing the suspension in the chip for more than 2 days.

value for a sphere and close to unity when the tactoid reaches extreme deformation, such as a long liquid filament. A linear trend is observed for the tactoids deformation parameter as a function of the dimensionless time, $\tau = t\dot{\varepsilon}_{xx}$, with $t$ the time. Similar behavior is also reported for simple viscoelastic fluids, when the stretching timescale is substantially lower than the time required for a droplet to reach steady-state deformation[29,30]. Thus, time-dependent deformation of tactoids follows $\mathcal{D} = (\partial\mathcal{D}/\partial\tau)\tau + \mathcal{D}_0$, where $\mathcal{D}_0$ captures the initial shape of the tactoids. In the Supplementary Note 6, we provide arguments allowing the estimation of $\mathcal{D}_0$ based on the equilibrium shape of the tactoids. As depicted in Fig. 3a–c, the transition between various classes of liquid crystalline tactoids is controlled by elongation, with bipolar and cholesteric tactoids turning into homogenous tactoids at a critical threshold. For the cholesteric to homogenous transition, there is a range of the deformation where the tactoids do not fall into a well-defined class, for example when the striped texture exists only partially along the tactoid. We indicate such a transient range with empty symbols in Fig. 3c.

Assuming that the tactoid geometry is axisymmetric with respect to its long axis during the elongation, the volume of the tactoids is measured and found to be almost constant (Fig. 3d–f), confirming that the tactoids undergo uniaxial extension during the deformation. However, there is slight decrease in the calculated volume for the cholesteric droplets during the deformation, which suggests that for this system the shape is not entirely axisymmetric. Essentially, the conditions for maintaining the axisymmetric geometry of the droplet depend on the thickness of the channel. In the case of channel height smaller

than the thickness of the droplet or so-called Hele-Shaw limit ($h < 2r$), the axisymmetric behavior will not be valid anymore and, as shown in the Supplementary Note 7, this is not the case in our study, confirming the axisymmetric geometry of the droplets during deformation.

To rationalize the effect of hydrodynamic forces on the elongation of the tactoids, we examined the deformation of the tactoids at different Capillary numbers, $Ca = \mu_I R_{eq.}\dot{\varepsilon}_{xx}/\gamma$, showing the ratio of viscous stretching forces $\mu_I R_{eq.}^2\dot{\varepsilon}_{xx}$ to the surface forces $\gamma R_{eq.}$ that resists the stretching, where $\mu_I$ is the viscosity of the continuous phase and $R_{eq.}$ is the equivalent radius of tactoids defined as[31]: $(r^2R)^{1/3}$, Fig. 3g–i. We measured $\mu_I$ to be 0.121 Pa s and $\gamma$ is calculated to be $1.1 \times 10^{-6}$ N m$^{-1}$, Supplementary Note 8. Inertial forces are neglected here as Reynolds number, Re $= \rho_I R_{eq.}U/\mu_I$, expressing the ratio of inertia to viscous forces, with $\rho_I$ is the density of the continuous phase, is very low and in the order of ~10$^{-7}$ (Supplementary Note 8). In the experiments, we kept the flow velocity lower than a critical value over which the fibrils in the isotropic phase align. Above such a flow velocity, that is well predicted by De Gennes[32,33], the medium becomes nematic and the induced birefringence of the medium blends with the one by the tactoids, making the tactoids undetectable. The continuous linear stretching behavior is observed at different Ca numbers for deformation parameter as a function of dimensionless time, i.e., $\mathcal{D} \sim \tau$. Such a continuous stretching and extreme deformation of the tactoids conceptually show that the very low interfacial and elastic forces of tactoids are not sufficient to balance the hydrodynamic force and keep the shape of the tactoids steady[34]. Having the relation of the $\partial\mathcal{D}/\partial\tau$ as a

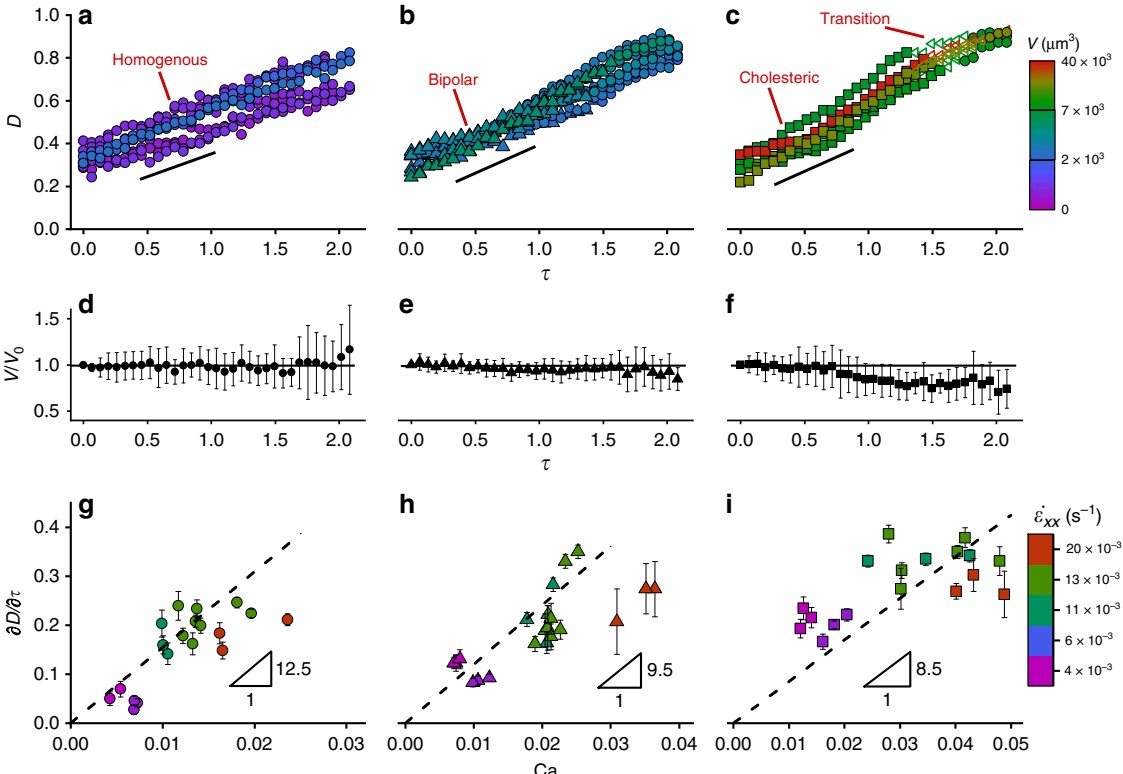

**Fig. 3 Deformation of different classes of liquid crystalline tactoids.** The data points presented by filled circle, triangle, and square correspond to homogenous, bipolar, and cholesteric tactoids, respectively. The empty symbols denote the transition regime of cholesteric to homogenous tactoids. **a–c** Deformation of tactoids under an extension rate of 0.013 s$^{-1}$ that shows a linear change in the deformation parameter $\mathcal{D}$ as a function of dimensionless time $\tau$ and order–order transitions of the bipolar and cholesteric tactoids to homogenous ones. **d–f** Variation in the volume of the tactoids under an extension rate of 0.013 s$^{-1}$ showing that the volume of the tactoids remains essentially constant. **g–i** The linear dependence of $\mathcal{D}$ on $\tau$ is tested for different Ca numbers. A linear fitting is used, as suggested by simple fluids theories. A total of 59 tactoids have been analyzed. The error bars in **d–i** denote standard deviation.

function of Ca number together with $\mathcal{D} = (\partial \mathcal{D}/\partial \tau)\tau + \mathcal{D}_0$, from earlier discussion, we obtain the following simple expression describing the tactoids time-dependent deformation:

$$\mathcal{D} = m\mathrm{Ca}\tau + \mathcal{D}_0, \qquad (1)$$

where $m$ is the slope of the dashed lines in Fig. 3g–i. Having $\mathcal{D}$ value from Eq. (1), one can get the aspect ratio of the tactoids as $\alpha = (1 + \mathcal{D})/(1 - \mathcal{D})$. Equation (1), that predicts the continuous deformation of the tactoids as a function of extension rate and the restoring effect of the interfacial tension, agrees well with the linear deformation concept considered for simple fluids when the stretching timescale of the droplets is significantly lower than their characteristic relaxation time[35–37]. Note that, although we assumed the same viscosity for all classes of tactoids, the viscosity is expected to decrease for the sequence cholesteric → bipolar → homogenous tactoids since the rods within the tactoids are more oriented toward the stretching direction in the homogenous configuration. Essentially, as it is also reflected in shear thinning behavior of liquid crystals (Supplementary Note 4), the viscosity of the liquid crystals decreases as the fibrils get oriented in the flow direction[38]. Thus, given the fact that a droplet with lower viscosity deforms easier due to less resisting droplet internal viscous forces during stretching[39], one can expect increasing deformations at a given extension rate for the sequence cholesteric → bipolar → homogenous. This behavior is reflected in the increase in $m$ value (Fig. 3g–i) in Eq. (1) when one progresses as cholesteric → bipolar → homogenous.

The change in $m$ value is related to the internal viscosity of the droplets since, for simple fluids droplet deformation under very low Reynolds number (as in this study), the deformation of the droplet depends on two dimensionless numbers: the capillary number and the ratio between the droplet internal viscosity and the medium viscosity, implying that both the internal viscosity of the droplet and the viscosity of the medium play a role in droplet deformation[36,37]. In Eq. (1), since the medium viscosity is the same for different classes of the tactoids, we relate the changes in $m$ for different classes of tactoids to their internal viscosities. The shape effect on $m$ can be ruled out based on the observation of the linear changes in the droplet deformation, featuring almost identical trends at different droplet elongation ratio. The elasticity effect on $m$ is also ruled out since for viscoelastic droplet transient deformation (similar to our study), negligible effects of elasticity on deformation have been reported in the literature[40]. Thus, Eq. (1) can be extended, in principle, to other liquid crystalline tactoids based on mesogens with different elasticity from that of amyloid fibrils.

**Order–order transitions in tactoids under deformation.** Having quantified the deformation of tactoids under extensional flow by Eq. (1), we now rationalize the director field transitions of liquid crystalline tactoids under deformation by looking at the free-energy landscape of the tactoids using a scaling form of Frank–Oseen elasticity theory. According to this theory, there are two energetic contributes associated with the tactoid at

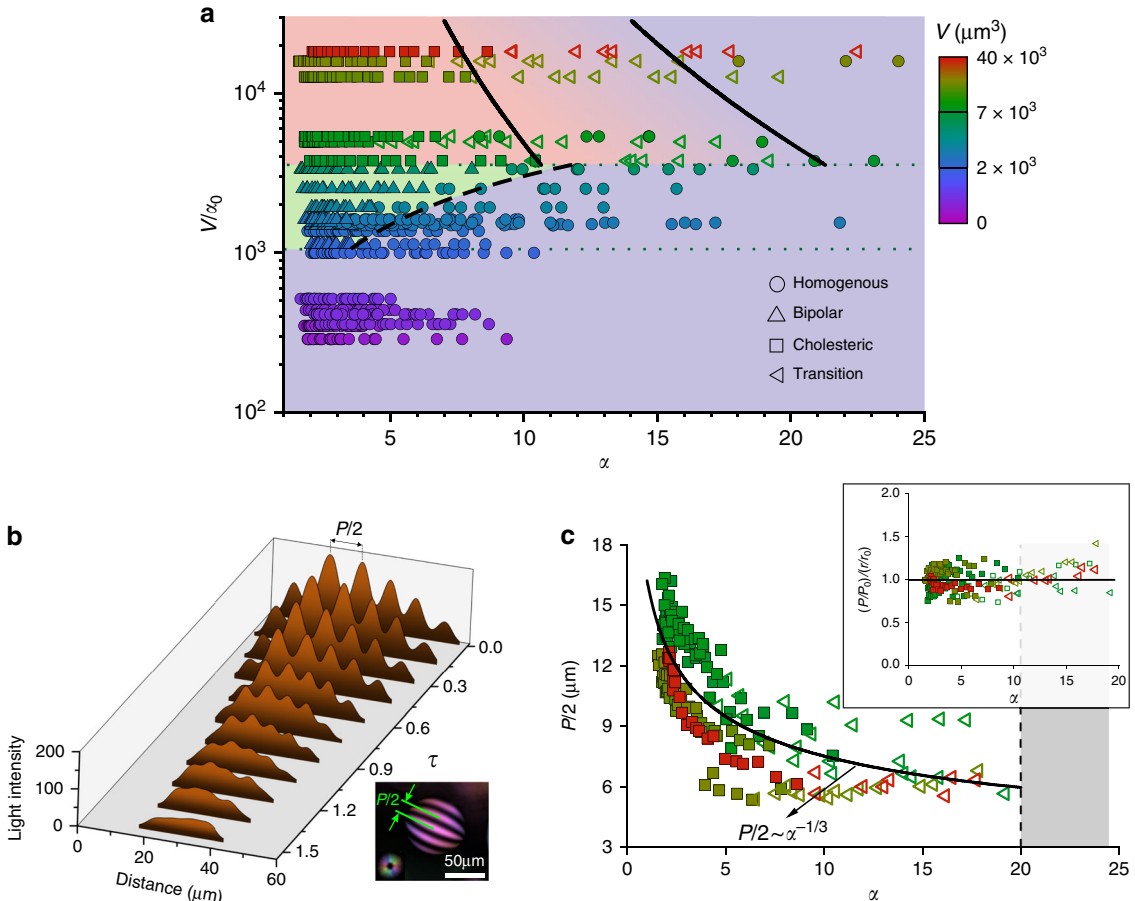

**Fig. 4 Phase diagram of nematic–cholesteric tactoids at different elongation ratios.** Filled circle, triangle, and square symbols denote homogenous, bipolar, and cholesteric tactoids. The empty symbols correspond to the transition regime of cholesteric to homogenous tactoids. **a** The developed theory predicts the transition for the bipolar to homogenous (dashed line), that is, $\alpha = cV\left(\frac{\gamma\omega}{K}\right)^3$ and cholesteric to homogenous (solid lines), that is, $\alpha = cV^{-1/5}\left(\frac{\gamma\omega}{K_2 q_\infty^2}\right)^{3/5}$. The term $\alpha_0$ denotes the initial aspect ratio of the tactoids. The constant $c$ value is 0.8 for bipolar to homogenous transition; and for cholesteric to homogenous tactoids, the $c$ value is 14.5 for the lower and 29.0 for the upper boundary of the transition region. **b** Evaluation of the cholesteric pitch under constant extension rate of 0.013 s$^{-1}$ showing both decrease of cholesteric pitch and order–order transition of the cholesteric tactoids. **c** Change in cholesteric pitch as a function of aspect ratio that scales as $P/2 \sim \alpha^{-1/3}$. The inset shows that ratio of $P/r$, for a given cholesteric tactoid, remains essentially constant at different elongation ratios.

equilibrium, namely the bulk elastic and surface free energies, and the total free energy of the tactoid can be expressed according to the following scaling form[3]:

$$F_E \sim \frac{Kr^2}{R} + \gamma Rr\left[1 + \omega(r/R)^2\right] + \frac{1}{2}K_2(\theta + q_\infty)^2 r^2 R, \quad (2)$$

where the first and last terms account for bulk elastic free energy and the middle term describes the surface free energy of the system. The importance of each term progressively shifts from left to right as the tactoids grow. Splay and bending free energy are embedded in the first term, where $K$ is the Frank elastic constant for splay and bending that are assumed to be equal here. The twist elastic free energy is given by the third term, where $K_2$ represents Frank twist elastic constant and $q_\infty$ is the chiral wave number, equal to $2\pi/P_\infty$ with $P_\infty$ the natural pitch of the system. The symbol $\theta$ represents the twist term equal to $n \cdot \nabla \times n$ in the Frank–Oseen elasticity theory with $n$ the nematic director. Finally, the middle term in Eq. (2) is the surface free energy of the system that accounts for both interfacial tension and anchoring strength, $\omega$. We measured and theoretically calculated the main constants entering in the free energy functional to be: $K = 1.0 \times$

$10^{-11}$ N, $K_2 = 2.0 \times 10^{-12}$ N, $P_\infty = 25.6$ μm, and $\omega = 1.17$ (Supplementary Notes 8 and 9).

To determine the critical aspect ratio at which the order–order transitions occur for a tactoid with a given volume, we examine the interplay between the terms in Eq. (2) upon transitions. The bipolar to homogenous transition happens when the bulk free energy of the bipolar tactoids, $\frac{Kr^2}{R}$, converts into the anchoring part of the surface free energy of the homogeneous tactoids, $\gamma Rr\omega(r/R)^2$. Thus, by setting free energies of the two phases equal at the transition, $\frac{Kr^2}{R} \approx \gamma Rr\omega(r/R)^2$ must hold, which can be reworked into:

$$\alpha|_{\text{bipolar}\to\text{homogenous}} \approx V\left(\frac{\gamma\omega}{K}\right)^3. \quad (3)$$

Using a similar approach, when the cholesteric to homogenous transition happens, the bulk free energy increases by $\frac{1}{2}K_2 q_\infty^2 r^2 R$ as $\theta$ changes from $-q_\infty$ to 0 in Eq. (2), while the surface free energy decreases by $C_i \gamma Rr\omega(r/R)^2$. Note that $\theta$ assumes the value of $-q_\infty$ for the cholesteric tactoids as the free energy of the cholesteric tactoids is in the lowest state when $\theta = -q_\infty$; and $\theta$ is zero for homogenous tactoids as there is no twist for the director field, i.e., $n \cdot \nabla \times n = 0$. The term $C_i$ is a constant that accounts for

the change in surface anchoring free energy when cholesteric changes to homogenous tactoids as the rods are anchored to the surface in different fashions in the two symmetries. Overall, from the equality of the free energies of the two phases at the transition, we have $C_i \gamma R r \omega (r/R)^2 \approx \frac{1}{2} K_2 q_\infty^2 r^2 R$, which upon simplifying and omitting pre-factors gives:

$$\alpha|_{\text{cholesteric}\rightarrow\text{homogenous}} \approx V^{-1/5} \left( \frac{\gamma \omega}{K_2 q_\infty^2} \right)^{3/5}. \quad (4)$$

Since Eqs. (3) and (4) are derived from a scaling form of the free energy functional, they are correct within a pre-factor. Yet, as can be seen by Fig. 4a, which brings together the scaling arguments and the experimental data providing the phase diagram of liquid crystalline tactoids at different elongation ratios, these scaling expressions capture very well the experimental findings and the corresponding order–order transitions. We argue here that as long as the elongation of the tactoids is uniaxial, one can take the theoretical framework presented in Fig. 4a as a universal phase diagram to predict the order–order transition of the tactoids under any type of the external force field.

To further elucidate the cholesteric droplet deformations and director field transitions, we measured the cholesteric pitch value under constant extension rate of 0.013 s$^{-1}$ as shown in Fig. 4b. Two main features can be noted from Fig. 4b: (i) the order–order transition of uniaxial cholesteric to homogenous tactoids that is observed by the decrease and annihilation of the cholesteric pitch and (ii) the number of cholesteric bands remains almost constant during the elongation, allowing us to state that the pitch and short axis of the cholesteric tactoids evolve proportionally, i.e., $P/2 \sim r$ (see the inset in Fig. 4c). This, together with the aforementioned discussion indicating that the volume of the droplet remains constant during elongation, i.e., $V \sim r^3 \alpha = $ constant, giving $r \sim \alpha^{-1/3}$, allows us to conclude that $P/2 \sim \alpha^{-1/3}$. Note that the low value of the twist elastic constant is understood to be the main responsible for the constant number of bands during tactoids elongation, setting to $\frac{1}{2} K_2 q_\infty^2 r^2 R$ the maximum free energy contribution related to twist and little resilience to changes in the twist angle needed to maintain the number of bands unaltered. Figure 4c presents the experimental data of the evolution of the pitch as a function of deformation, showing a good agreement with the expected scaling law. Such a change in cholesteric pitch value, for instance from $P/2 = 15\,\mu\text{m}$ to $P/2 = 6\,\mu\text{m}$ for a tactoid of $V = 7500\,\mu\text{m}^3$, demonstrates the possibility of tuning the cholesteric pitch and hence the wavelength of light that is transmitted/reflected by them[41]; however, practically, this is limited to the cases where the stretching of the cholesteric tactoids using extensional flow or other external force field becomes feasible. Figure 4c does not allow to draw conclusions on the pitch evolution beyond $P/2 = 6\,\mu\text{m}$ (range highlighted in gray); however, it is inferred that in order to go from $P/2 = 6\,\mu\text{m}$ to infinite, as should be for homogeneous tactoids, a sudden jump must be expected, which would point to a first-order thermodynamic transition. While this cannot be conclusively assessed here, to rule out the possibility of a plausible, but not observable, pitch beyond $P/2 = 6\,\mu\text{m}$ in Fig. 4c, we checked the images resolution, and found that the resolution being around six times finer than the minimum observed pitch value, a continuous evolution of the pitch beyond $P/2 = 6\,\mu\text{m}$ can be ruled out, which re-inforces the expected first-order nature of the transition. Additionally, the deformed cholesteric tactoids that have undergone an order–order transition to homogenous tactoids, in the contraction zone, were checked at the beginning of the relaxation (right after contraction zone, see Fig. 2c) and no sign of pitch was found. This, together with the smooth evolution of the pitch in the observable window (Fig. 4b), further discards the possibility of

a continuous decrease of pitch beyond $P/2 = 6\,\mu\text{m}$, reinforcing the argument on the first-order thermodynamic nature of this transition.

To conclude, we have shown that using microfluidic to produce a purely extensional flow can be used as a mean to induce alignment, deformation, order–order transitions, coalescence, and break-up of amyloid-based liquid crystalline tactoids. Under such an imposed extensional flow, the tactoids are shown to undergo extreme deformations at a Capillary number as low as $\sim 10^{-2}$. By combining experiments and scaling arguments on the free energy functional, we have been able to rationalize the threshold at which order–order transitions are expected and to debate on the thermodynamic nature of these transitions. Our results open a new window on confinement-induced features of colloidal systems at extremely small interfacial tension and can pave the way to new strategies in the design of self-assembled complex fluids.

## Methods

**Preparation of amyloid fibrils suspension.** First, we dispersed 6 g of β-lacto-globulin, purified from whey protein (see ref. [42] for details), in 300 ml of Milli-Q water that was set to pH 2 by adding HCl after dispersion, giving 2 w/w of β-lactoglobulin/Milli-Q water. This was followed by filtration of the dispersion through a 0.45-μm Nylon syringe filter (Huberlab) to remove any possible aggregation. Then, following heat-induced denaturation process, we let the solution over a hot plate (IKA, RCT basic) for 5 h at 90 °C, while the magnetic stirrer (length: 3 cm and diameter: 0.6 cm) was rotating at ~1000 r.p.m. inside the suspension to avoid forming of any gel close to the air interface.

The amyloid fibrils in the suspension were cut by applying mechanical shear force, giving the length distribution reported in the Supplementary Note 1. To clear the suspension from any unreacted monomers and peptides, it was dialyzed for 5 days using 100 kDa (MWCO) Spectra/Por dialysis membrane (Biotech CE Tubing) against 10 L Milli-Q water, adjusted to pH 2, with everyday bath change. The desired concentration for the suspension was adjusted by up-concentrating it through reverse osmosis using 6–8 kDa (MWCO) Spectra/Por 1 dialysis membrane (Standard RC Tubing) against 10 wt% polyethylene glycol solution (mol wt: $M_r$ ~ 35,000, Sigma Aldrich) that was set to pH 2. The end concentration used in the experiments was 2.2 wt%, that is in the biphasic region, with 2.0 wt% and 2.5 wt% concentrations for isotropic and nematic phases, respectively, which were measured after the macroscopic phase separation occurred. Note that, after every step, the suspension was checked between crossed polarizers for any gelation or larger protein aggregates. The concentration measurement was made gravimetrically using a Mettler AT20 microbalance.

**Characterization of the physical properties of the amyloid fibrils.** A suspension of 0.01 wt% was prepared from a 2.2 wt% amyloid fibrils suspension used in the experiments by diluting it in pH 2 Milli-Q water. As described elsewhere[1], to perform the atomic force microscopy (AFM) measurement, the amyloid fibrils solution was deposited on a freshly cleaved mica by placing a droplet of the prepared suspension for 120 s. This is followed by rinsing the mica with Milli-Q water and drying it with pressurized air. MultiMode VIII scanning probe microscope (Bruker) was used to acquire the images, while running in tapping mode at ambient conditions. The images acquired in AFM were analyzed by FiberApp[43] allowing to track each fiber and measure the length and height distribution, respectively.

**Microfluidic.** To fabricate the microfluidic chip, the standard soft lithography procedure was followed[44]. The microfluidic channel was made by plasma-bonding of the PDMS channel to plain glass slides (Corning 2947). The attached PDMS channel was made from a 10:1 mixture of polydimethylsiloxane (PDMS) monomer and curing agent (Dow Corning Slygard 184).

The used microfluidic chip had rectangular cross section. The four sets of geometrical dimensions used in this study are: (1) $w_u = 600\,\mu\text{m}$, $w_t = 50\,\mu\text{m}$, and $h = 100\,\mu\text{m}$; (2) $w_u = 1000\,\mu\text{m}$, $w_t = 200\,\mu\text{m}$, and $h = 200\,\mu\text{m}$; (3) $w_u = 1000\,\mu\text{m}$, $w_t = 150\,\mu\text{m}$, and $h = 200\,\mu\text{m}$; (4) $w_u = 1000\,\mu\text{m}$, $w_t = 100\,\mu\text{m}$, and $h = 200\,\mu\text{m}$. The length of the contraction zone was kept the same for all geometrical dimensions at $l_c = 1000\,\mu\text{m}$, while different entrance lengths ($l_e$) were used, 1500, 3000, and 4500 μm.

**Sample characterization.** To identify the symmetry of the tactoids, polarized optical microscopy Zeiss with attached camera (AxioCam MRc) was used. Unless otherwise noted, objectives 5× (Achrostigmat) and 10× (Plan Neofluar) were used to capture the images between crossed polarizers. For the cases of under the flow field measurement, time series acquisition mode at a frame rate of 12 frames per minute was used. The acquired images were analyzed using ImageJ. For quantitative analysis of tactoid deformation, the microfluidic channel was mounted on

the microscope keeping the flow direction at 45° with respect to one of the crossed polarizers, while to support the order–order transition analysis the microfluidic channel was rotated at different angles with respect to crossed polarizers, see Supplementary Note 2 for more details.

**Experimental detail**. All of the tests were run at room temperature. The suspension was filled in a 250-μl syringe (Hamilton) and pumped at different flow rates using a syringe pump (Harvard Apparatus). For the external fluidic interconnects, short pieces of a flexible tubing with an inner diameter 0.8 mm was used. To connect the tubing to the microfluidic channel, a needle with inner and outer diameters of 0.34 mm and 0.64 mm, respectively, was used. A new microfluidic chip was used upon significant deposition of the suspension on the channel wall to avoid any disturbance in the flow field.

**Rheological measurements**. To measure the viscosity of the liquid crystalline phases, an MCR 502 (Anton Paar) rheometer with PP25 parallel plate geometry and 1 mm gap was used. Due to possible errors associated with the measurement of low viscosities in this geometry, the validity of the measurement was checked using a mixture of water–glycerol with known viscosity that was in the same range of liquid crystal phases viscosities. All of the measurements were completed at room temperature, similar to the conditions at which the experiments are performed. To avoid possible evaporation of the suspension during the measurement, a solvent trap was used in all the measurements.

## Data availability

The data that support the findings of this study are available from the corresponding author upon request.

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

## Acknowledgements

We thank J. Adamcik for the atomic force microscopy measurements, X. Cao (ETHZ) for assistance in fabrication of the microfluidic chips, and P. Fischer (ETHZ) and M. Diener for valuable discussions. Prof. Andrew de Mello (ETHZ) is kindly acknowledged for granting access to his laboratory to fabricate microfluidic chips. We thank P. Azzari for valuable discussions and sharing the results of his under-preparation study in the rebuttal letter of this work.

## Author contributions

H.A. designed and performed the experiments, contributed to the theoretical scaling, analyzed the data, interpreted the results, and wrote the manuscript. M.A. contributed to the experiments. R.M. designed and directed the research, analyzed the data, interpreted the results, developed the theoretical formalism, and wrote the manuscript. All authors discussed the results and commented on the manuscript.

## Competing interests

The authors declare no competing interests.
