## [Peer Review File · Nature Communications]

REVIEWER COMMENTS

Reviewer #1 (Remarks to the Author):

Please see the attached review

Reviewer #2 (Remarks to the Author):

The paper presents a very interesting study of the flow induced transitions in amyloid fibril liquid crystalline tactoids. The leading author with his group started the field of amyloid fibril based nano colloidal lyotropic liquid crystals several years ago. Most of their previous studies were devoted to near equilibrium structures of nematic and cholesteric tactoids with low surface tension formed in water based solvents. In this study extensional viscous flow, induced force was used to produce extreme deformations of tactoids. Results are clearly explained. Although the paper is well written, there are some points that need to be improved before the paper would be ready for publication in Nature Communications. Below I list my remarks.

19

The description "elongated oblate shapes" is confusing for prolate shapes.

31-36

When introducing microfluidic devices that form monodispersed emulsions it would be good to cite also D. Weitz who introduced the device and later use it for nematic droplets.

69 and later in the text and SI

The use of the term "height" to characterize lateral size of amyloid fibrils has sense for somebody who studies fibrils on a solid substrate with the AFM, but it definitely is not appropriate for the NC audience trying to understand what fibrils dispersed in aqueous solvent do.

Are the mean lengths of fibrils really determined up to four digits? It would be good to mention in the main text also the spread of dispersions.

89

Here the lower case u is used for the velocity and later for the same quantity the capital U is used. Unfortunately, the capital U is used also for free energy in Eq.2.

94-95

It would be good, that beside flow rates and extensional rates, also velocities are listed!

95-96

In addition, thickness of the channel should be specified!

115

What happens with the flow after leaving the contraction part? Is there a vortex or more of them? This can explain rotation. How much and how fast velocity is reduced?

124-129

The behavior after the contraction, particularly coalescence and formation of bulk cholesteric are just briefly mentioned related to Fig 2. Probably this will appear in the next paper, but this should be stated.

172

Axisymmetric geometry probably depends on the thickness of the channel. This should be commented. Can this be examined by a side view (y direction) observation of tactoids?

178-179

It would be helpful for the readers to write separately "viscous stretching forces" and "surface forces" in

addition to their ratio in the form of the Capillary number.

179 and 183

Two different symbols are used to represent viscosity!

206

"Resulting in larger deformation" should be more precisely explained as one can first think that the friction force is decreased.

228

The symbol "theta" should be explained as Eq.2 is here simply copied from a previous paper.

238-252

In this, segment there are several confusing statements. Instead of the term "energy", "free energy" should be used. At phase transitions, the free energies of the two related phases are equal. Simply claiming "energy conservation" could lead to a miss understanding although it is true that the free energies of the two phases after the restructuring are the same.

I have another question about bipolar – homogenous transition. As anchoring of the nematic is probably not very strong, the director angle on the tactoid surface deviates from tangential already in the original bipolar structure and can with extension of the tactoid continuously transform to the axial direction. Is the change in texture clearly supporting the abrupt change?

256-258

For examining, more details of structural transitions it would be helpful if one can instead of abrupt increase in the channel cross section continuously (symmetrically) expand its cross section back to the original value.

264-266

Why pitch follows the change of a lateral dimension of a tactoid should be explained. Is the low value of the twist elastic constant the main reason?

269-270

If the claim "...demonstrates the possibility of tuning the cholesteric pitch and hence the wavelength of light that is transmitted/reflected by ..." is targeting applications, it is probably too ambitious as the change occurs only in the extensional flow.

287-288

The claim "By combining theory and experiments" is too strong as authors use only simplified scaling based approaches.

294-305

In fig 4 the difference in the volume color-coding of squares standing for the cholesteric phase in Segments a and c is confusing!

The cholesteric squares corresponding to $\alpha > 10$ appearing in Segment c do not appear in the Segment a!

Do the existence of cholesteric squares above natural pitch demonstrates that because of the weak elastic constant the pitch can be stretched as well?

Reviewer #3 (Remarks to the Author):

In this work the authors describe the observed experimentally effects of an extensional flow field in deforming and through the deformation possibly inducing a phase change of tactoids originally developed with a homogeneous nematic, biaxial nematic and cholesteric liquid crystalline states using carefully prepared solutions of amyloid fibrils. They show how simple arguments based on the energetics of the liquid

crystalline states can be used to deduce the conditions applicable at the observed phase transitions; simple extensional deformation calculations can similarly be invoked to explain the observed changes in the pitch of cholesterics as a function of the deformation.

This work represents a significant extension of previous under equilibrium conditions work of the authors of amyloid fibril liquid crystalline tactoids, under flow deformation. In particular, they show how one can exploit the extensional field as generated in a suitably constructed microfluidic device to deform the tactoids and possibly induce a phase change to the liquid crystalline structure. The accompanying analysis nicely explains the observed phenomena. However, there are a few issues, as discussed in the detailed comments below, that I feel the authors need to adequately respond in a (minor) revision before their work can be recommended for publication.

Detailed comments

1. Perhaps, in the introduction, along with the studies mentioned on amyloid fibril liquid crystalline tactoids, especially the recent study by the authors on the effects of fibril lengths (reference 1, provided) the authors can also mention (as they discuss further effects on the cholesteric pitch of the deformation) the recent study by Wensink on the effects of size polydispersity on the pitch of cholesterics (reference [1] cited below).
2. The much higher surface tension of the oil droplets used in Fig. 2f (as compared to the tactoids) should be underlined and mentioned also in that Figure caption (corresponding to a much higher Capillary number) in addition of discussing it in the text.
3. In the paragraph after Eq. (1), lines 200 – 207, arguments are made to justify the observed changes in m value between the different liquid crystalline structures based on changes in viscosity observed due to shear thinning associated with different liquid crystalline phases (cases g, h, i in figure 3). However, those changes affect the internal viscosity within the tactoids and in the deformation of the tactoids the viscosity that enters is that of the external phase which is supposed to be the same. In addition, in all these cases the extensional rate was the same. My feeling is that those changes can be better justified through changes in the volume and shape of the tactoids, as well as in their elasticity.
4. After Eq. (2) can you also explain what θ is in that equation? Further down (line 246) you mention the values that it can take—can you also explain why?

5. Some Editorial Remarks

- a) In line 5, use “follows” instead of “follow”
- b) In line 218, use “remains” instead of “remain”
- c) In line 219, use “is tested” instead of “are tested”
- d) In figure 4, can you explain what $\alpha_{>0}$ is? (equilibrium aspect ratio?)
- e) In line 348, can you also add the symbol for the entrance length?
- f) The year of publication for reference 1 should be 2019 not 2018.

References

[1] H.H. Wensink, Effect of Size Polydispersity on the Pitch of Nanorod Cholesterics, Crystals 2019, 9, 143; doi:10.3390/cryst9030143.

Reviewer #1

Below is our response to Reviewer 1.

The manuscript entitled, “Flow-Induced order-order transitions in amyloid fibril liquid crystalline tactoids” deals with the biphasic phase of lyotropic liquid crystals for by amyloid fibrils. The authors have designed a microfluidic channel that is capable of creating a flow field that would be elongational in nature. They then proceed to deform the biphasic phase (which is composed of nematic, and cholesteric droplets in an isotropic matrix) and find that the director configuration changes upon the system experiencing the flow field. Just to be sure, this phase (in fact calling this a “phase” is an error in itself, but I will do so fully recognizing that it is wrong!) in itself is in the metastable state, as if the biphasic material is looked at infinite time, there will be two-phases with one interface, and a meniscus. Hence, to some extent using terms like “symmetry breaking”, “order-order” transitions are quite problematic, whether it is explicit or implicit. The authors have previously published a number of papers dealing with configurational transitions of the droplets and have termed these director configuration changes as “order-order transitions”, and rather implicitly suggesting that they are analogous to “symmetry breaking” phase transitions. At the very outset, I want to make it clear that this kind of language in papers is very disingenuous, and must not ever be used. Just to be clear, the transitions that the authors talk deal with changes in the director configuration when confined, either to spherical objects or ellipsoidal objects, and many of these are often referred to as tactoids – So, much of the study deals with the flow behavior and more precisely the director configuration changes due to the imposed flow field. I would not use the term “Order-Order transitions” to describe such director configuration changes – It is perhaps “sexy” to use the term order-order transitions but this should not be the case. For example, in the case of block copolymers, one can, in fact have order-order transitions due to flow fields as the materials transitions through the various ordered phases (I would encourage the authors to look up papers by Takei Hashimoto in the late 1980’s dealing with order-order transitions). These are sort of the general comments at this point after having the read the paper many times over, just to be sure of my impression. I will try to outline my specific comments below, and some of them will deal with the general comments as well.

First, we should clarify that the term “biphasic phase” is used (several times) only by the Reviewer and not by us, as we find this terminology incorrect (biphasic implies two phases, not one); Indeed later the Reviewer acknowledges recognizing that it is wrong, still using it. In the manuscript, the terms “biphasic region” and “biphasic regime” are instead used which is a very common terminology in the liquid crystals community and can be found in authors as eminent as Paul J. Flory (see for example: Selected Works of P.J. Floy, Volume III, Part 7: Liquid Crystals). See later for more comments on this point.

With respect to the terminology order-order transitions and symmetry breaking we must respectfully, yet strongly disagree with the referee.

To start, the main ordered phases on amyloid liquid crystalline tactoids are homogeneous, bipolar, uniaxial cholesteric and radial cholesteric and they all correspond to a relative minimum of the free energy of the systems, described by the Frank-Oseen Hamiltonian combined with the anisotropic surface tension energetic contribution as described by Rapini-Papoular. As such, since the different phases stem out from a minimization of the total free energy and since they are all characterized by a different order parameter, this makes already appropriate to discuss transitions among them as order-order transitions or phase transitions (see also our reply to the same referee further below). With respect to the terminology of “symmetry breaking”, inferred by the Reviewer, we actually do not use this phrasing at all! We use solely the word “symmetry” and only in the context of the discussion of the symmetry of the nematic director field in the various tactoids (which is obviously different). Even then, we shall note that the term “symmetry breaking” is a terminology widely accepted in literature among most respectable scientists from the liquid crystal community (see for example: Tortora & Lavrentoich “Chiral symmetry breaking by spatial confinement in tactoidal droplets of lyotropic chromonic liquid crystals”, PNAS, 2011, or similarly Prinsen P, van der Schoot P (2004) “Parity (*for symmetry*) breaking in nematic tactoids”. J Phys Condens Matter 16:8835–8850. We also shall remark that based on the above, even the term symmetry breaking would be appropriate since we do observe a “*Chiral symmetry breaking*” when going from cholesteric to homogeneous tactoids. So, in short, although we do not use the phrasing symmetry breaking at all, this would still be appropriate in the present context.

Secondly, we are able to instruct the referee even with the precise nature of thermodynamic phase transition occurring for each order-order transition. To support the use of the terms “order-order transition” and “different symmetry” (also from the last comment of the Reviewer), we provide the Reviewer below with the results from our group on the evolution of total free energy as a function of volume of the tactoid. The calculation of the total free energy E is based on the Frank-Oseen-Rapini-Papoular energetic description of the tactoids computed by a variational theory approach. The referee can find the details in Bagnani et al. Sci. Rep. 2019. To better catch the transitions, the total free energy E has been reduced by the isotropic surface free energy of a sphere of equivalent volume, E_0 , which is a well-behaved function without singularities. Therefore, any singularity in the evolution of $E-E_0$ reveals the nature of the thermodynamic phase transitions (see Figure 1 here below). As the Referee can observe the derivative (i.e. the slope) of the free energy vs volume changes continuously across the homogeneous-bipolar transition and discontinuously across the bipolar-uniaxial cholesteric and uniaxial cholesteric-radial cholesteric transition. According to Ehrenfest classification of phase transitions, this allows identifying a second (or higher) order transition when going from homogenous to bipolar, followed by two first order transitions from bipolar→uniaxial cholesteric and uniaxial cholesteric→onion-like radial cholesteric. In agreement with this picture, the Reviewer can also consult

Figure S5 in Nystrom et al. Nature Nanotechnology 2018, where he/she will be able to see that while coexistence of bipolar and uniaxial tactoids is observed across a large range of tactoid volumes (as it should be for a first order transition), no coexistence is observed for homogeneous-bipolar tactoids, as expected for a second order thermodynamic transition. The physical origins of these discontinuities are discussed below when replying further to the referee on the nature of the phase transitions among different tactoids. Here, however, we limit ourselves to observe that both theory and experiments fully justify the terminology order-order (or phase) transitions.

Fig. 1. Free energy landscape of different class of tactoids. The color coding of the background is as: blue: homogeneous, green: bipolar, yellow: uniaxial cholesteric, red: onion-like radial cholesteric. In y axis, $E_0 = \gamma 4\pi V^{2/3}$, i.e. the isotropic surface free energy of a sphere of volume V , where V is volume quoted in the x axis.

In the abstract the authors state that “Liquid crystalline droplets, also known as tactoids,, challenge our current understanding of liquid crystals under confinement”. This is such a vague statement and rather generic to make in an abstract – I would have liked to see what the challenges are, and how those challenges are addressed – the authors point out that much of the studies are done under quiescent conditions, and they are so for a good reason. This paper does describe experiments done in the biphasic region and provides the results of flow fields on the director configuration of droplets, and of course based on the flow field the director configuration changes. Not a surprise!

The Reviewer points that in the abstract we should state the challenges in the field of liquid crystalline droplets and how these challenges are addressed. We believe that content-wise we followed the Nature Communications guidelines on abstract “Abstract: Provide a general introduction to the topic and a brief nontechnical summary of your main results and their implication.” Furthermore, the total length of the abstract should be kept to less than 150 words despite this general introduction. However, we take note

of this comment from the Reviewer in modifying the abstract of the work explaining what the main challenges are (complex non-spherical shapes and lack of knowledge in the dynamic properties):

The modified abstract is as follows:

“Liquid crystalline droplets, also known as tactoids, forming by nucleation and growth within the phase diagram region where isotropic and nematic phases coexist, challenge our understanding of liquid crystals under confinement, due to anisotropic surface boundaries at vanishingly small interfacial tension, resulting in complex, non-spherical shapes. Little is known about their dynamical properties, since they are mostly studied under quiescent, quasi-equilibrium conditions. Here we show that different classes of amyloid based nematic and cholesteric tactoids undergo order-order transitions by flow-induced deformations of their shape. Tactoids align under extensional flow, undergoing extreme deformation into highly elongated prolate shapes, with the cholesteric pitch decreasing as an inverse power-law of the tactoids aspect ratio. Free energy functional theory and experimental measurements are combined to rationalize the critical elongation above which the director-field configuration of tactoids transforms from bipolar and uniaxial cholesteric to homogenous and to debate on the thermodynamic nature of these transitions.”

We need to respectfully disagree with respect to the statement “Not a surprise”. We indeed believe that the current study is surprising from several perspectives, some of which are listed in the following:

1. The experimental behavior of liquid crystalline droplets under extensional flow is itself challenging, unexplored and surprising! This stems out from the fragile nature of liquid crystalline tactoids combining vanishingly small interfacial tension, nematic field and mass exchange between the isotropic and the nematic phase during the nucleation and grow process. This report is to the best of our knowledge the very first study on dynamic study of tactoids and the assessment of shape transitions under flow.
2. The section of the text related to Fig. 2 describes completely novel results for the liquid crystalline droplets under microfluidic conditions. For instance, it is surprising that using the presented channel geometry the tactoids rotate by 90° , including cholesteric droplets that have aspect ratio close to unity. This opens interesting perspectives even on simpler fluid droplets: for instance it is possible that the same rotation behavior could also happen in simple fluid droplet in the literature but, due to lack of internal structure in these fluids, the rotation could have remained undiscovered to date.

3. It is surprising that liquid crystalline tactoids show similar hydrodynamic behavior to that of simple fluids under uniaxial extensional flow, connecting the understanding of simple fluid droplet hydrodynamics to that of liquid crystalline droplets. This is discussed in detail in the section related to Figure 3 of the manuscript.
4. The discovery of the transition of tactoids under an extensional flow that occurs when the tactoids are deformed beyond a critical threshold (Fig. 4) is surprising. This is so, as discussed in point 1 above, since the common expectation could be that the internal configuration of the tactoids is disturbed already at very small deformation. However, we see in the current study that the tactoids including the cholesteric ones maintain an internal configuration up to very high aspect ratios, allowing a continuous decrease of their cholesteric pitch (Fig. 4). It should be noted that the transition of cholesteric to homogenous is itself a surprising finding as for the equilibrium condition the transition follows a homogenous→bipolar→cholesteric sequence, while under the extensional flow it follows a cholesteric → homogenous sequence. There is a whole section in SI of the current work only devoted to correctly capture this type of transition.

The paper starts out by noting that “Liquid crystalline droplets or tactoids form through self-organization of mesogens in concentrated aqueous suspensions (ref. 1,2), and are characterized by vanishingly low interfacial tensions”. There are a few problems with the statement as it stands: First of all reference 1,2, in my opinion (and I am sure the authors will disagree), are not appropriate – this is largely because they are not the first ones to observe such droplets. While they do eventually cite JD Bernal and company, I would argue those are the references that should be presented first, if we want to be scientifically honest. Second problem with the statement concerns the use of the word “mesogens” and it often refers to mesogenic units (that are rodlike) in a polymer or those units themselves forming a liquid crystalline phase. I have hardly ever encountered the use of mesogen for the classical rodlike polymer, Tobacco Mosaic Virus (TMV), a favorite object of study for physicists, largely due to its hard-rod character, meaning the persistence length being greater than the contour length. It is important to be accurate in describing things, and I find that often in the manuscript many terms, are used, rather casually. Very discouraging!

We cite References 1 and 2 at the beginning of the manuscript because they are the most relevant papers to the content of the current manuscript. Ref. 1 is directly related because it is one of the few works in the literature about amyloid fibrils liquid crystalline droplets, i.e. the same system studied in the current work. Ref. 2 by van der Schoot et al. is a crucial work as it describes the scaling form of part of the Frank–Oseen elasticity theory (excluding chiral twist) used to predict homogenous-bipolar transition. We fully acknowledge all of the previous efforts in scientific community and we did our best to cite the most relevant works, albeit following a narrative and not a historical order. For example, as also the

Reviewer pointed out, when talking about the discovery of liquid crystalline tactoids in dispersions of biological rod-like systems, Ref. 12-15 were cited, which to the best of our knowledge reflect the contributions of those individuals who made such discoveries.

Besides, in other parts of the manuscript we cite G. I. Taylor as a the first person who studied droplet deformation (Ref. 26 in the main text), G. H. McKinley as the first group who introduced extensional flow in the microfluidic chip (Ref. 21), D. A. Weitz as a pioneer in introducing water-water emulsion in microfluidic chip (Ref. 20), G. M. Whitesides who introduced rapid fabrication of the microfluidic chip as the same method used here (Ref. 41). Later, we cite De Gennes (Ref. 30), L. G. Leal (Ref 25 and 28), and H. A. Stone (Ref. 4) due to their pioneering studies related to the field.

Furthermore, the Reviewer argues that he/she hardly ever encountered the use of mesogen for classical rod-like colloidal objects such as Tobacco Mosaic Virus. In fact, we must respectfully disagree. We used the term “mesogen” to refer for the objects that form the liquid crystalline droplets in the main text and this is common practice in the literature. To provide a support, in the following we list several books/papers that use the term “mesogen” to indicate rod-like colloidal particles that form liquid crystals:

1. D. Demus, J. Goodby, G. W. Gray, H.-W. Spiess and V. Vill, Handbook of Liquid Crystals, Wiley-VCH, Weinheim, Vol. 3, (1998).

This is a book with 5 authors and more than 30 contributors and on page 8:

*“A further point concerning the first phase of our history of liquid crystals is about nomenclature, a matter about which scientists of today still love to argue. ... Friedel preferred the term mesomorphic to describe the liquid crystal state, and the associated term mesophase, reflecting the intermediate nature of these phases between the crystalline and isotropic liquid states. These terms are again widely used today and coexist happily with the Lehmann terminology. A useful term springing from Friedel’s nomenclature is the word **mesogen** (and also nematogen and smectogen), used to describe a material that is able to produce mesophases.”*

* Friedel, G., Les états mésomorphes de la matière. In Annales de physique Vol. 9 (1922).

2. Wang, P. X., & MacLachlan, M. J.. Liquid crystalline tactoids: ordered structure, defective coalescence and evolution in confined geometries. *Philosophical Transactions of the Royal Society A: Mathematical, Physical and Engineering Sciences* 376, 20170042 (2018).

3. Wang, P. X., Hamad, W. Y., & MacLachlan, M. J. Structure and transformation of tactoids in cellulose nanocrystal suspensions. *Nature communications*, 7, 1-8 (2016).

Similar to Ref. 1, MacLachlan et al. used the term mesogen in the above works to indicate ribbon-like or rod-like objects that show liquid crystalline behavior. From paper No. 2 above:

“Nematic tactoids formed in lyotropic liquid crystals of ribbon-like or rod-like **mesogens**, such as vanadium pentoxide (figure 1a), aluminium oxyhydroxide (figure 1b) [18], **tobacco mosaic viruses** (figure 1c) or carbon nanotubes (figure 1d) [19], are prolate spindle-shaped or spheroidal microdroplets with circular arc boundaries (figure 1a)....In a study by Hirai et al. [11] on phase-separated **cellulose nanocrystal** suspensions, the rod-shaped **mesogens** in the liquid crystalline phase were found to have significantly greater lengths and higher aspect ratios than those in the isotropic phase. ”

4. Weirich, K. L., Banerjee, S., Dasbiswas, K., Witten, T. A., Vaikuntanathan, S., & Gardel, M. L. Liquid behavior of cross-linked actin bundles. *Proceedings of the National Academy of Sciences*, 114, 2131-2136 (2017).

Stating that: “The transition from homogeneous to bipolar depends on the ratio of the length of the **mesogen** and the tactoid length (L); the transition has recently been demonstrated with carbon nanotube tactoids”

5. Jampani, V. S. R., Volpe, R. H., de Sousa, K. R., Machado, J. F., Yakacki, C. M., & Lagerwall, J. P. F. Liquid crystal elastomer shell actuators with negative order parameter. *Science advances*, 5(4), eaaw2476 (2019).

We read in the paper:

“For example, liquid crystals (LCs) are nonsolid condensed phases with long-range orientational order, giving them anisotropic physical properties such as optical birefringence. In conventional LCs, the constituents, referred to as **mesogens** (they can be individual molecules, aggregates such as micelles, or **nanoparticles**), align with their principal axis along a common symmetry axis, dubbed the director, nStudied between crossed polarizers, a nematic formed by rod-shaped **mesogens** with maximum electronic polarizability along the long axis shows positive uniaxial birefringence with n as its optic axis, i.e., the refractive index is greater along n than perpendicular to it”

There are many other examples which could be quoted (*Chirality in Liquid Crystals*, edited by Heinz Kitzerow, Christian Bahr or again *Nematic and Cholesteric Liquid Crystals: Concepts and Physical Properties* By Patrick Oswald, Pawel Pieranski) but we think we have provided enough examples already.

We finally note that the persistence length of these fibrils is of the order of 2 microns (see Nystron et al Nature Nanotechnology 2018 or Bagnani et al ACS Nano 2019), that is roughly one order of magnitude above the average contour length of the fibrils, i.e. fully justifying a rod-like behavior, hence the terminology of mesogen for the amyloid fibrils.

Then in the next paragraph, the author's state "...within the isotropic (I) plus nematic (N) biphasic regime..." – this is at best confused terminology, not something I would expect from a group that consider to be excellent! Then, later in the paragraph the authors point to their discovery of "amyloid fibrils show transition between six different equilibrium symmetries of the nematic field configuration..." - These are all various director configurations of nematic or cholesterics when confined to spherical cavities - The use of the word "symmetries" is troublesome to say the least. Then much later in the paragraph the authors use "symmetry" and the term flow-induced order-order transition – both of these are used in the context of symmetry breaking "phase transitions". These changes in director configurations are NOT phase transitions and symmetry of the phase, the nematic phase has not changed! There are many places where this sort of thing is used so often that it is in fact very detracting from the manuscript. The observations of the droplet deformation and the like are nice, but the basic premise of the paper that talks about it from the perspective of order-order transitions (implying phase transitions) is just a flawed argument. There are many other issues with the manuscript but I shall not enunciate them here, only to say that the premise is just incorrect.

The Reviewer pointed that he/she is confused by "...within the isotropic (I) plus nematic (N) biphasic regime...", However the reason is not provided. We really do not understand the issue as the terminology "biphasic" has been largely accepted from Flory to current time and we can re-iterate this is correct. Just to avoid any potential confusion and as a service to the reader, we change the text to "...where isotropic (I) and nematic (N) phases coexist¹⁻³".

For our reply to the second section of the comment, we address the Reviewer to our earlier reply to his/her first comment. We must however briefly note that the referee is unfortunately wrong in the present case since the observed structural changes are effectively phase transitions, whose thermodynamic nature we have already discussed above. The Reviewer would be correct if the changes of the nematic field of different nematic phases would occur in pure bulk, where the changes in free energy and order parameter would be smooth curves without singularities. In the dispersed (tactoid) state, however, the picture is very different: the anchoring contribution of the surface free energy of the tactoids undergoes the following discontinuous evolution *finite*→*zero*→*finite*→*zero* when undergoing the sequence *homogenous*→*bipolar*→*uniaxial cholesteric*→*radial cholesteric*, introducing singularities. This discontinuous behavior is at the origin of the thermodynamic transitions discussed above.

After spending an enormous amount of time going through a number of papers published by the authors as well as many of the references provided, and after considerable reflection, I cannot honestly say that this should be published in the pages of Nature Communications.

For all the reasons quote above, we must respectfully disagree with the assessment of this Reviewer and have provided extensive arguments on why the terminology we use is correct.

Reviewer #2

We would like to thank the Reviewer #2 for his/her comments on our manuscript. Below is our response with references to the pages of the revised manuscript.

The paper presents a very interesting study of the flow induced transitions in amyloid fibril liquid crystalline tactoids. The leading author with his group started the field of amyloid fibril based nano colloidal lyotropic liquid crystals several years ago. Most of their previous studies were devoted to near equilibrium structures of nematic and cholesteric tactoids with low surface tension formed in water based solvents. In this study extensional viscous flow, induced force was used to produce extreme deformations of tactoids. Results are clerly explained. Although the paper is well written, there are some points that need to be improved before the paper would be ready for publication in Nature Communications. Bellow I list my remarks.

19: The description “elongated oblate shapes” is confusing for prolate shapes.

We thank the Reviewer for pointing out this issue. In the revised manuscript, we changed the “elongated oblate shape” to the “elongated prolate shapes” in abstract as:

“Tactoids align under extensional flow, undergoing extreme deformation into highly elongated prolate shapes, with the cholesteric pitch decreasing as an inverse power law of the tactoids aspect ratio.”

31-36: When introducing microfluidic devices that form monodispersed emulsions it would be good to cite also D. Weitz who introduced the device and later use it for nematic droplets.

We agree with the Reviewer and, in the revised manuscript, we cited the following work from D. Weitz in the first paragraph of the Introduction on page 2. As pointed by the Reviewer, in this work, D. Weitz *et al.* introduce a device to produce highly monodisperse micron-sized water-oil and the droplets of liquid crystalline phase in another immiscible liquid with interfacial tension in the order of water-oil emulsion.

7. Umbanhowar, P. B., Prasad, V. & Weitz, D. A. Monodisperse emulsion generation via drop break off in a coflowing stream. *Langmuir* **16**, 347–351 (2000).

In the revised manuscript, we cited the above work as Ref. 7 and the text in first paragraph of the introduction is modified to:

“While microfluidic has been widely used to study water-oil emulsions⁴⁻⁷ and to produce monodisperse micron-sized droplets of liquid crystals in another immiscible liquid with interfacial tension in the order of water-oil emulsion⁷, the hydrodynamics of liquid crystalline tactoids remains poorly studied since most of the studies in these systems aimed at understanding primarily their thermodynamically equilibrium features⁸.”

69: and later in the text and SI, the use of the term “height” to characterize lateral size of amyloid fibrils has sense for somebody who studies fibrils on a solid substrate with the AFM, but it definitely is not appropriate for the NC audience trying to understand what fibrils dispersed in aqueous solvent do. Are the mean lengths of fibrils really determined up to four digits? It would be good to mention in the main text also the spread of dispersions.

Our response is categorized as three bullets in the following:

-We agree with the Reviewer and used term “diameter (D_f)” to denote the lateral size of the fibrils in the revised version.

-The fibrils length is measured using FiberApp open source software developed in our group allowing us to report the length and diameter up to six digits. However, the number of the digits reported by software is merely based on the image processing calculation (e.g. number of pixels per fibril length) and is not supported by the resolution of the AFM instrument. Taking into account the resolution of our AFM instrument that is, respectively, 1-2 nm and 0.1 nm for length and lateral size measurements, we can state that the mean lateral size and length of the fibrils can be determined up to one decimal digit and zero decimal digit, respectively. Thus, in the revised manuscript, we modified the text and reported the mean lengths and lateral size of the fibrils as $L_{f,m}=303$ nm and $D_{f,m}=2.5$ nm, respectively (see below).

- As Reviewer pointed, we added the spread of dispersions in the main text of the revised manuscript on page 3 as following:

“The mean length of fibrils is $L_{f,m}=303$ nm with $44 \leq L_f \leq 1500$ nm and standard deviation in normal distribution ± 194 nm and lognormal distribution fitting parameters $\mu_{fitting} = 5.5 \pm 0$ and $\sigma_{fitting} = 0.6 \pm 0$; the mean height of the fibrils which corresponds to the fibrils average diameter considering a cylindrical approximation is $D_{f,m}=2.5$ nm, with $1.0 \leq D_f \leq 6.2$ nm and standard deviation in normal distribution ± 0.7 nm and lognormal distribution fitting parameters $\mu_{fitting} = 0.9 \pm 0$ and $\sigma_{fitting} = 0.3 \pm 0$, see Fig. S1.”

89: Here the lower case u is used for the velocity and later for the same quantity the capital U is used. Unfortunately, the capital U is used also for free energy in Eq.2.

We thank the Reviewer for pointing to this out. In the revised manuscript: (i) the term u_x shows the flow speed (ii) the term U denotes the flow speed in the straight channel before and after contraction zone ($U=Q/[w_uh]=Q/[w_dh]$, where w_d is the width of downstream channel that is equal to the width of upstream channel w_u for all used geometries), (iii) We also modified Eq. 2 where now F_E denotes the free energy, following the nomenclature used by Prinsen and van der Schoot (Ref. 2 in the main text).

While u_x is already defined as $u_x=Q/[w(x)h]$ in the main text, to clearly define U as well, we added on page 4:

“The flow speed in the straight channel before and after the contraction zone is $U=Q/[w_uh]=Q/[w_dh]$, where w_d is the width of downstream channel that is equal to w_u for all used geometries, and was always found in the range of $0.49 \leq U \leq 1.92 \mu\text{m s}^{-1}$ in the experiments.”

We also modified Eq. 2 to:

$$“ \quad F_E \sim \frac{Kr^2}{R} + \gamma Rr[1 + \omega(r/R)^2] + \frac{1}{2}K_2(\theta + q_\infty)^2 r^2 R \quad (2) ”$$

94-95: It would be good, that beside flow rates and extensional rates, also velocities are listed!

We agree with the Reviewer and, in the revised manuscript, the velocities are listed. We added on page 4:

“The flow speed in the straight channel before and after the contraction zone is $U=Q/[w_uh]=Q/[w_dh]$, where w_d is the width of downstream channel that is equal to w_u for all used geometries, and was always found in the range of $0.49 \leq U \leq 1.92 \mu\text{m s}^{-1}$ in the experiments.”

95-96: In addition, thickness of the channel should be specified!

We specified the thickness of the channel in the revised manuscript. We added on page 4:

“The microfluidic channel was fabricated with two different heights (or thicknesses) h , equal to $100 \mu\text{m}$ and $200 \mu\text{m}$, see Methods for detailed information.”

115: What happens with the flow after leaving the contraction part? Is there a vortex or more of them? This can explain rotation. How much and how fast velocity is reduced?

To provide a comprehensive reply to the Reviewer comment, we performed a simulation using ANSYS Fluent software in identical flow speed and channel geometry to that of Figure 2, i.e. $U= 1.5 \mu\text{m s}^{-1}$, $w_u=w_d = 600 \mu\text{m}$, $w_t = 50 \mu\text{m}$, and $h = 100 \mu\text{m}$ (see following Fig. S3). The simulation was performed

for an isotropic non-Newtonian fluid having the same properties of the isotropic phase of our system. The results show that there is no vortex formation after fluid leaves the contraction part, Fig. S3b. In addition, we are able to see how the velocity decreases from contraction part to downstream section. Theoretically, the average flow speed should decrease by the inverse ratio of the channel width, i.e. $U/u_t = w_t/w_d$ as volumetric flow rate Q is constant for a given flow condition, u_t shows the flow speed at the throat section of the contraction part. Thus, considering that in Fig. 2, the channel geometry is $w_d = 600 \mu\text{m}$ and $w_t = 50 \mu\text{m}$, the average flow speed is expected to decrease by 1/12 as $U/u_t = w_t/w_d = 1/12$.

To provide a reasoning on the rotation of the tactoids, we also measured the extension rate of the flow in y direction in the downstream channel $\dot{\epsilon}_{yy} = \partial u_y / \partial y$ and compared it to the shear rate in the flow direction $\dot{\Gamma}_{xy} = \partial u_x / \partial y$, see Fig. S3c to d. Essentially, as it is well explained in Ref. 22 (in the revised manuscript 24) of the main text, when the $\dot{\epsilon}_{yy}$ is strong compared to $|\dot{\Gamma}_{xy}|$, (i.e. $\dot{\epsilon}_{yy}/|\dot{\Gamma}_{xy}| > 0.14$), cylindrical particles rotate by $\sim 90^\circ$, aligning their long axes perpendicular to flow direction. Our results show that in the center of channel right after the contraction zone the extension rate $\dot{\epsilon}_{yy}$ is higher than shear rate $|\dot{\Gamma}_{xy}|$, explaining the mechanism behind the rotation of the tactoids shown in Fig. 2. In fact, stronger extension rate $\dot{\epsilon}_{yy}$ when compared to $|\dot{\Gamma}_{xy}|$ forces the tactoids to align their long axis in the direction of $\dot{\epsilon}_{yy}$. It can also be seen that as we move away from the throat the values of $\dot{\epsilon}_{yy}$ and $|\dot{\Gamma}_{xy}|$ get closer to each other, Fig. S3c to d.

We added the above information along with Fig. S3 to the SI:

“III. Tactoids rotation in expansion zone

To investigate the rotation of the tactoids, we performed a simulation using ANSYS Fluent software in identical flow speed and channel geometry to that of Figure 2, i.e. $U = 1.5 \mu\text{m s}^{-1}$, $w_u = w_d = 600 \mu\text{m}$, $w_t = 50 \mu\text{m}$, and $h = 100 \mu\text{m}$ (Fig. S3). The simulation was performed for an isotropic non-Newtonian fluid having the same properties of the isotropic phase of our system. The results show that there is no vortex formation after fluid leaves the contraction part, Fig. S3b. In addition, we are able to see how the velocity decreases from contraction part to downstream section, Fig. S3a. Theoretically, the average flow speed should decrease by the inverse ratio of the channel width, i.e. $U/u_t = w_t/w_d$ as volumetric flow rate Q is constant for a given flow condition, u_t shows the flow speed at the throat section of the contraction part. Thus, considering that in Fig. 2, the channel geometry is $w_d = 600 \mu\text{m}$ and $w_t = 50 \mu\text{m}$, the average flow speed is expected to decrease by 1/12 as $U/u_t = w_t/w_d = 1/12$.

We also monitored the extension rate of the flow in y direction in the downstream channel $\dot{\epsilon}_{yy} = \partial u_y / \partial y$ and compared it to the shear rate in the flow direction $\dot{\Gamma}_{xy} = \partial u_x / \partial y$, see Fig. S3c to d. Essentially, as it is well explained in Ref. 3, when the $\dot{\epsilon}_{yy}$ is strong compared to $|\dot{\Gamma}_{xy}|$, (i.e. $\dot{\epsilon}_{yy}/|\dot{\Gamma}_{xy}| > 0.14$), cylindrical particles rotate by $\sim 90^\circ$, aligning their long axes perpendicular to flow direction. Our results show that in the center of channel right after the contraction zone the extension

rate $\dot{\epsilon}_{yy}$ is higher than shear rate $|\dot{\Gamma}_{xy}|$, explaining the mechanism behind the rotation of the tactoids shown in Fig. 2. In fact, stronger extension rate $\dot{\epsilon}_{yy}$ when compared to $|\dot{\Gamma}_{xy}|$ forces the tactoids to align their long axis in the direction of $\dot{\epsilon}_{yy}$. It can also be seen that as we move away from the throat the values of $\dot{\epsilon}_{yy}$ and $|\dot{\Gamma}_{xy}|$ get closer to each other, Fig. S3c to d.

Fig. S3 | Simulation results of the fluid flow in expansion section of the used microfluidic channel. a, Velocity magnitude counter. b, Streamlines colored by flow speed magnitude. c-e, Comparison of $\dot{\epsilon}_{yy}$ (gray line) and $|\dot{\Gamma}_{xy}|$ (orange line) at different distance from throat in the channel, where the distances from throat are 10 (c), 50 (d), and 100 (e) μm .

“

124-129: The behavior after the contraction, particularly coalescence and formation of bulk cholesteric are just briefly mentioned related to Fig 2. Probably this will appear in the next paper, but this should be stated.

We agree with the Reviewer and, in the revised manuscript on page 6, we address this issue by adding:

“The detailed discussion on coalescence and the formation of uniform bulk cholesteric phase (Fig. 2i) is beyond the scope of this paper and will be addressed in a separate detailed investigation.”

172: Axisymmetric geometry probably depends on the thickness of the channel. This should be commented. Can this be examined by a side view (y direction) observation of tactoids?

The Reviewer is correct and the axisymmetric geometry of the droplet depends on the thickness of the channel. In the case of channel height smaller than the thickness of the droplet or so-called Hele-Shaw limit ($h < 2r$), the axisymmetric behavior will not be valid anymore (which is not the case in our study, as discussed below).

Regarding the observation of the side view of the channel, we believe that this would be a very interesting experiment to perform. However, based on our several experimental trials, it is almost impossible experimentally to detect the channel (and subsequently the tactoids) from side view under microscope due to (i) the high PDMS thickness from the side view and (ii) the optical properties of PDMS resulting from the cutting of the PDMS. The volume calculations (in the main text) and the comparison of droplet thickness and channel height (see below), make us confident to claim the axisymmetric geometry for the tactoids, regardless of the inaccessible side view observation.

To clarify that the axisymmetric geometry of the droplet depends also on the thickness of the channel, in the revised manuscript we added on page 8:

“Essentially, the conditions for maintaining the axisymmetric geometry of the droplet depend on the thickness of the channel. In the case of channel height smaller than the thickness of the droplet or so-called Hele-Shaw limit ($h < 2r$), the axisymmetric behavior will not be valid anymore and, as is shown in the SI, this is not the case in our study, confirming the axisymmetric geometry of the droplets during deformation.”

And, in the SI, we added:

“

VIII. Tactoids short axis evolution versus channel height and width

Figure S8a and b show respectively the evaluation of the $2r/h$ and $2r/w(x)$ for all the different classes of tactoids at different extensional rate during deformation along the contraction zone. For the vast majority of the tactoids studied in this work, the $2r/h$ and $2r/w(x)$ values are less than 0.3 confirming that we are far from Hele-Shaw limit for droplet or $2r < h$ and $2r < w(x)$ where axisymmetric geometry for the droplet does no longer stand valid.

Fig. S8 | Tactoids short axis evolution versus channel height and width. Evaluation of the $2r/h$ (a) and $2r/w(x)$ (b) for different classes of tactoids considered in this study at different extensional rate during deformation along the contraction zone.

“

178-179: It would be helpful for the readers to write separately “viscous stretching forces” and “surface forces” in addition to their ratio in the form of the Capillary number.

We agree with the Reviewer and, in the revised manuscript, we implemented this point by modifying the text on page 8 as following:

“To rationalize the effect of hydrodynamic forces on the elongation of the tactoids, we examined the deformation of the tactoids at different Capillary numbers $Ca = \mu_1 R_{eq} \dot{\epsilon}_{xx} / \gamma$, showing the ratio of viscous stretching forces $\mu_1 R_{eq}^2 \dot{\epsilon}_{xx}$ to the surface forces γR_{eq} that resists the stretching, where μ_1 is the viscosity of the continuous phase and R_{eq} is the equivalent radius of tactoids defined as:³¹ $(r^2 R)^{1/3}$, Fig. 3g-i.”

179 and 183: Two different symbols are used to represent viscosity!

We thank the Reviewer for pointing this mistake, which was a typo in the manuscript. In the revised manuscript, we made sure that we used μ_1 to show the viscosity of the continuous phase (isotropic phase) and μ_N to denote the viscosity of the droplet phase (nematic phase). Additionally, ρ_I , showing the density of the isotropic phase, is corrected in the revised manuscript on page 9.

206: “Resulting in larger deformation” should be more precisely explained as one can first think that the friction force is decreased.

We modified the text and provided precise explanation on larger deformation in the revised manuscript as following:

“Note that, although we assumed the same viscosity for all classes of tactoids, the viscosity is expected to decrease for the sequence cholesteric → bipolar → homogenous tactoids since the rods within the tactoids are more oriented toward the stretching direction in the homogenous configuration. Essentially, as it is also reflected in shear thinning behavior of liquid crystals (see SI), the viscosity of liquid crystals decreases as the fibrils get oriented in the flow direction³⁸. Thus, given the fact that a droplet with lower viscosity deforms easier due to less resisting droplet internal viscous forces during stretching³⁹, one can expect increasing deformations at a given extension rate for the sequence cholesteric → bipolar → homogenous. This behavior is reflected in the increase in m value (Fig. 3g-i) in Eq. 1 when one progresses as cholesteric → bipolar → homogenous.”

39. Delaby, I., Muller, R., & Ernst, B. Drop deformation during elongational flow in blends of viscoelastic fluids. Small deformation theory and comparison with experimental results. *Rheol. Acta* **34**, 525-533 (1995).

228: The symbol “theta” should be explained as Eq.2 is here simply copied from a previous paper.

In the revised manuscript, we explained “theta” on page 12 by adding the following:

“The symbol θ represents the twist term equal to $n \cdot \nabla \times n$ in the Frank–Oseen elasticity theory with n the nematic director.”

238-252: In this, segment there are several confusing statements. Instead of the term “energy”, “free energy” should be used. At phase transitions, the free energies of the two related phases are equal. Simply claiming “energy conservation” could lead to a miss understanding although it is true that the free energies of the two phases after the restructuring are the same.

We thank the Reviewer for pointing out this. In the revised manuscript, we made sure we used the term “free energy” instead of the energy (see the text in the following).

In addition, to avoid the misunderstanding by using the term “energy conservation” as pointed by the Reviewer, we changed the text to “at the transition the free energies of the two phases must be equal”.

We implemented the above changes in the revised manuscript. On page 12-13, we changed the text to:

“To determine the critical aspect ratio at which the order-order transitions occurs for a tactoid with a given volume, we examine the interplay between the terms in Eq. 2 upon transitions. The bipolar to

homogenous transition happens when the bulk free energy of the bipolar tactoids, $\frac{Kr^2}{R}$, converts into the anchoring part of the surface free energy of the homogeneous tactoids, $\gamma Rr\omega(r/R)^2$. Thus, by setting free energies of the two phases equal at the transition, $\frac{Kr^2}{R} \approx \gamma Rr\omega(r/R)^2$ must hold, which can be reworked into:

$$\alpha|_{\text{bipolar} \rightarrow \text{homogenous}} \approx V \left(\frac{\gamma\omega}{K} \right)^3 \quad (3)$$

Using a similar approach, when the cholesteric to homogenous transition happens, the bulk free energy increases by $\frac{1}{2}K_2q_\infty^2r^2R$ as θ changes from $-q_\infty$ to 0 in Eq. 2, while the surface free energy decrease by $C_i\gamma Rr\omega(r/R)^2$. Note that θ assumes the value of the $-q_\infty$ for the cholesteric tactoids as the free energy of the cholesteric tactoids is in the lowest state when $\theta=-q_\infty$; and θ is zero for homogenous tactoids as there is no twist for the director field, i.e. $n \cdot \nabla \times n = 0$. The term C_i is a constant that accounts for the change in surface anchoring free energy when cholesteric changes to homogenous tactoids as the rods are anchored to the surface in different fashions in the two symmetries. Overall, from equality of the free energies of the two phases at the transition, we have $C_i\gamma Rr\omega(r/R)^2 \approx \frac{1}{2}K_2q_\infty^2r^2R$, which upon simplifying and omitting pre-factors gives:

$$\alpha|_{\text{cholesteric} \rightarrow \text{homogenous}} \approx V^{-1/5} \left(\frac{\gamma\omega}{K_2q_\infty^2} \right)^{3/5} \quad (4)$$

“

I have another question about bipolar – homogenous transition. As anchoring of the nematic is probably not very strong, the director angle on the tactoid surface deviates from tangential already in the original bipolar structure and can with extension of the tactoid continuously transform to the axial direction. Is the change in texture clearly supporting the abrupt change?

The change in the texture is continuous if we consider the bipolar tactoids at time zero and see how it evolves during the elongation. As the Reviewer correctly pointed out, we observe that the tactoids continuously elongates and the internal configuration of the tactoids adapts itself to the new configuration (see Fig. S2). During such elongation as the curvature of the interface decreases the internal configuration continuously transform to the axial direction. This change happens continuously until a critical aspect ratio is reached, above which the tactoids cannot hold anymore the bipolar

configuration and the transition to homogenous structure takes place. Thus, we can state that the whole process of the transition from bipolar tactoid at rest to homogenous is continuous transition. This is also supported by the reply arguments provided to reviewer 1. However, from an experimental point of view, the tactoids configuration can be distinguished as bipolar and homogenous, respectively below and above the critical aspect ratio, so that we can state that the transition happens in proximity of such a critical aspect ratio. Whether the dynamic of the process changes the nature of this transition from second order to first order is a fascinating point that is very well taken and will be considered with high priority in our future work. This will require a robust implementation of the energetic terms related to the flow in the Frank-Oseen Hamiltonian and an accurate calculation of the free energy in proximity of this transition. We sincerely thank the referee for this inspiring comment!

As a note, given the anchoring of our system $\omega = 1.17$ being higher than one, the system can be characterized by high anchoring: see Ref. 18 in the revised manuscript where onion-like radial cholesteric transition is induced by anchoring in amyloid fibrils system.

To implement the point in the revised manuscript, we added on SI that:

“Note that the tactoids continuously elongates and the internal configuration of the tactoids adapts itself to the new configuration (see Fig. S2). During such elongation as the curvature of the interface decreases the internal configuration continuously transforms in the axial direction. This happens until a critical aspect ratio is reached, above which the tactoids cannot hold anymore the bipolar configuration and the transition to homogenous structure takes place. Thus, we can state that the whole process of the transition from bipolar tactoid at rest to homogenous is a continuous transition. However, from an experimental point of view, the tactoids configuration becomes well distinguishable as bipolar and homogenous, respectively below and above the critical aspect ratio, so that we can state that the transition happens in proximity of such a critical aspect ratio.”

256-258: For examining, more details of structural transitions it would be helpful if one can instead of abrupt increase in the channel cross section continuously (symmetrically) expand its cross section back to the original value.

Indeed, this is very interesting idea by the Reviewer and we expect that using this geometry, apart from learning the details of structural transitions, will result in quite novel behavior of liquid crystalline droplets. A recent work by Anke Linder et al.^a published in Nature Physics shows that achiral straight flexible filaments travelling in the microfluidic channel geometry suggested by the Reviewer, change their shape to helicoidal. Thus, we believe that an independent work should focus on the behaviors of liquid crystalline droplets in the expansion phase and we indeed are planning to conduct such investigation. It is anticipated that if the relaxation time of the droplet (governed by droplet properties)

is higher than the droplet retraction time scale by compressional flow, similar complex behavior to that of Ref. *a* below could be expected. However, to be accurate with these claims, a systematic and focused work will be needed, which is well beyond the scope of this manuscript.

- a. Chakrabarti, Brato, Yanan Liu, John LaGrone, Ricardo Cortez, Lisa Fauci, Olivia du Roure, David Saintillan, and Anke Lindner. "Flexible filaments buckle into helicoidal shapes in strong compressional flows." *Nature Physics* (2020): 1-6.

264-266: Why pitch follows the change of a lateral dimension of a tactoid should be explained. Is the low value of the twist elastic constant the main reason?

Quantitatively, the pitch follows the change of a lateral dimension of the tactoid as the number of cholesteric bands remains essentially constant during the tactoids elongation. We believe that, as correctly pointed out by the Reviwer, the low value of the twist elastic constant is the main reason for this observation. This is reflected the bulk free energy term $\frac{1}{2}K_2q_\infty^2r^2R$ that increases when going from cholesteric to homogenous, setting $\frac{1}{2}K_2q_\infty^2r^2R \approx C_i\gamma Rr\omega(r/R)^2$ at transition, showing that low value for the twist elastic constant can delay the transition and hence the cholesteric pitch will be present in the tactoids up to higher elongation ratio.

We added above explanation to the revised manuscript on page 13 as:

“Note that the low value of the twist elastic constant is understood to be the main responsible for the constant number of bands during tactoids elongation, setting to $\frac{1}{2}K_2q_\infty^2r^2R$ the maximum free energy contribution related to twist and thus offering little resilience to changes in the twist angle needed to maintain unaltered the number of bands.”

269-270: If the claim “...demonstrates the possibility of tuning the cholesteric pitch and hence the wavelength of light that is transmitted/reflected by ...” is targeting applications, it is probably too ambitious as the change occurs only in the extensional flow.

To clarify this issue, we changed the text on page 13 to:

“Such a change in cholesteric pitch value, for instance from $P/2=15 \mu\text{m}$ to $P/2=6 \mu\text{m}$ for a tactoid of $V=7500 \mu\text{m}^3$, demonstrates the possibility of tuning the cholesteric pitch and hence the wavelength of light that is transmitted/reflected by them⁴¹; however, practically, this is limited to the cases where the stretching of the cholesteric tactoids using extensional flow or other external force field becomes feasible.”

287-288: The claim “By combining theory and experiments” is too strong as authors use only simplified scaling based approaches.

In the revised manuscript on page 14, we modified the wording to:

“By combining experiments and scaling arguments on the free energy functional, we have been able to rationalize the threshold at which order-order transitions are expected and to debate on the thermodynamic nature of these transitions.”

294-305: In fig 4 the difference in the volume color-coding of squares standing for the cholesteric phase in Segments a and c is confusing! The cholesteric squares corresponding to $\alpha > 10$ appearing in Segment c do not appear in the Segment a! Do the existence of cholesteric squares above natural pitch demonstrates that because of the weak elastic constant the pith can be stretched as well?

We modified the color-coding in Segment c of the Fig. 4 and made it the same as segment a to avoid confusion (see following).

The existence of the cholesteric pitch above natural pitch, as explained in one of the comments before and correctly pointed out by the Reviewer, could be related to the low value of the twist elastic constant. This is reflected in the Eq. 4 or the bulk free energy term $\frac{1}{2}K_2q_\infty^2r^2R$ that increases when going from cholesteric to homogenous, setting $\frac{1}{2}K_2q_\infty^2r^2R \approx C_i\gamma Rr\omega(r/R)^2$ at transition, showing that low value for the twist elastic constant can delay the transition and hence the cholesteric pitch will be present in the tactoids during the tactoids elongation.

We added above explanation to the revised manuscript on page 13 as:

“Note that the low value of the twist elastic constant is understood to be the main responsible for the constant number of bands during tactoids elongation, setting to $\frac{1}{2}K_2q_\infty^2r^2R$ the maximum free energy contribution related to twist and thus offering little resilience to changes in the twist angle needed to maintain unaltered the number of bands.”

The Figure 4 is modified to:

Fig. 4 | Phase diagram of nematic–cholesteric tactoids undergoing order-order transitions induced by an extensional flow and evaluation of cholesteric pitch at different elongation ratios. Filled circle, triangle, and square symbols denote homogenous, bipolar, and cholesteric tactoids. The empty symbols correspond to transition regime of cholesteric to homogenous tactoids. **a**, The developed theory predict the transition of the bipolar to homogenous (dashed line) and cholesteric to homogenous (solid lines) tactoids. The term α_0 denotes the initial aspect ratio of the tactoids, see SI for the estimation of α_0 based on the equilibrium aspect ratio of the tactoids. The constant c value is 0.8 for bipolar to homogenous transition; and for cholesteric to homogenous tactoids, the c value is 14.5 for lower and 29.0 for upper boundary of transition region. **b**, Evaluation of the cholesteric pitch under constant extension rate of 0.013 s^{-1} showing both decrease of cholesteric pitch and order-order transition of the cholesteric tactoids. **c**, Change in cholesteric pitch as a function of elongation ratio that scales as $P/2 \sim \alpha^{-1/3}$. The inset shows that ratio of P/r , for a given cholesteric tactoid, remains essentially constant at different elongation ratios.

Reviewer #3

We would like to thank the Reviewer #3 for his/her comments on our manuscript. Below is our response with references to the pages of the revised manuscript.

In this work the authors describe the observed experimentally effects of an extensional flow field in deforming and through the deformation possibly inducing a phase change of tactoids originally developed with a homogeneous nematic, biaxial nematic and cholesteric liquid crystalline states using carefully prepared solutions of amyloid fibrils. They show how simple arguments based on the energetics of the liquid crystalline states can be used to deduce the conditions applicable at the observed phase transitions; simple extensional deformation calculations can similarly be invoked to explain the observed changes in the pitch of cholesterics as a function of the deformation.

This work represents a significant extension of previous under equilibrium conditions work of the authors of amyloid fibril liquid crystalline tactoids, under flow deformation. In particular, they show how one can exploit the extensional field as generated in a suitably constructed microfluidic device to deform the tactoids and possibly induce a phase change to the liquid crystalline structure. The accompanying analysis nicely explains the observed phenomena. However, there are a few issues, as discussed in the detailed comments below, that I feel the authors need to adequately respond in a (minor) revision before their work can be recommended for publication.

Detailed comments

1. Perhaps, in the introduction, along with the studies mentioned on amyloid fibril liquid crystalline tactoids, especially the recent study by the authors on the effects of fibril lengths (reference 1, provided) the authors can also mention (as they discuss further effects on the cholesteric pitch of the deformation) the recent study by Wensink on the effects of size polydispersity on the pitch of cholesterics (reference [1] cited below: [1] H.H. Wensink, Effect of Size Polydispersity on the Pitch of Nanorod Cholesterics, *Crystals* 2019, 9, 143; doi:10.3390/cryst9030143.).

As the Reviewer pointed, in the revised manuscript, we added following text to the Introduction and mentioned the interesting work of H. H. Wesink.

“It is also important to mention recent findings on the effects of the length and polydispersity of mesogens on the cholesteric pitch, showing that the pitch decreases with increasing amyloid fibrils length¹ while length polydispersity of mesogens enhances the twist elastic modulus²¹.”

21. Wensink, H. H. Effect of size polydispersity on the pitch of nanorod cholesterics. *Crystals* **9**, 143 (2019)

2. The much higher surface tension of the oil droplets used in Fig. 2f (as compared to the tactoids) should be underlined and mentioned also in that Figure caption (corresponding to a much higher Capillary number) in addition of discussing it in the text.

We agree with the Reviewer and, in the revised manuscript, we modified the text related to panel f of Figure 2 to highlight the much higher surface tension of the oil droplets as following:

“A control experiment on oil-in-water droplets with interfacial tension in the order of $\sim 0.01 \text{ N m}^{-1}$, i.e. orders of magnitudes higher than the liquid crystalline tactoids, illustrate how the oil droplet remains essentially undeformed through the same flow field conditions used to deform the tactoids.”

3. In the paragraph after Eq. (1), lines 200 – 207, arguments are made to justify the observed changes in m value between the different liquid crystalline structures based on changes in viscosity observed due to shear thinning associated with different liquid crystalline phases (cases g, h, i in figure 3). However, those changes affect the internal viscosity within the tactoids and in the deformation of the tactoids the viscosity that enters is that of the external phase which is supposed to be the same. In addition, in all these cases the extensional rate was the same. My feeling is that those changes can be better justified through changes in the volume and shape of the tactoids, as well as in their elasticity.

We thank the Reviewer for pointing this out and to clarify the issue we sharpened the text and explained in details why the different values of m are related to the internal viscosity of the tactoids. We explained that, for the simple fluids droplet and under very low Reynolds numbers (as here in our study), it is shown that the deformation of the droplet depends on two non-dimensional numbers: capillary number and the ratio of droplet internal viscosity to medium viscosity implying that both the internal viscosity of the droplet and the viscosity of the medium play role in droplet deformation (Reference 36 and 37 in the revised manuscript by R. G. Cox and J. M. Rallison, respectively). We clarified that a droplet with lower viscosity deforms easier due to lower resisting droplet internal viscous forces during stretching against external viscous stretching forces (Ref. 39 in the revised manuscript). In Eq. 1 (that captures the capillary number), the medium viscosity is considered through the capillary number and, as also mentioned by the Reviewer, it is the same for different classes of the tactoids, so we relate the changes in m for different classes of tactoids to their internal viscosities.

The changes in m is not related to the volume as it is captured in capillary number by radius. The shape effect is ruled out based on the observation of the linear changes in the droplet deformation showing almost identical trend at different droplet elongation ratio. Finally, the elasticity is also ruled out since for viscoelastic droplet transient deformation (similar to our study), significant effect by elasticity on

deformation is not reported in the literature (Re. 40 in the revised manuscript). Thus, considering the above, we believe that internal viscosities of the tactoids should be the main affecting parameter on m values.

39. Delaby, I., Muller, R., & Ernst, B. Drop deformation during elongational flow in blends of viscoelastic fluids. Small deformation theory and comparison with experimental results. *Rheol. Acta* **34**, 525-533 (1995).

We modified the text in the revised manuscript on page 9 to address the Reviewer comment as following:

“Note that, although we assumed the same viscosity for all classes of tactoids, the viscosity is expected to decrease for the sequence cholesteric \rightarrow bipolar \rightarrow homogenous tactoids since the rods within the tactoids are more oriented toward the stretching direction in the homogenous configuration. Essentially, as it is also reflected in shear thinning behavior of liquid crystals (see SI), the viscosity of liquid crystals decreases as the fibrils get oriented in the flow direction³⁸. Thus, given the fact that a droplet with lower viscosity deforms easier due to less resisting droplet internal viscous forces during stretching³⁹, one can expect increasing deformations at a given extension rate for the sequence cholesteric \rightarrow bipolar \rightarrow homogenous. This behavior is reflected in the increase in m value (Fig. 3g-i) in Eq. 1 when one progresses as cholesteric \rightarrow bipolar \rightarrow homogenous.

The change in m value is related to the internal viscosity of the droplets since, for simple fluids droplet deformation under very low Reynolds number (as in this study), the deformation of the droplet depends on two dimensionless numbers: the capillary number and the ratio between droplet internal viscosity and the medium viscosity, implying that both the internal viscosity of the droplet and the viscosity of the medium play a role in droplet deformation^{36,37}. In Eq. 1, since the medium viscosity is the same for different classes of the tactoids, we relate the changes in m for different classes of tactoids to their internal viscosities. The shape effect on m can be ruled out based on the observation of the linear changes in the droplet deformation, featuring almost identical trends at different droplet elongation ratio. The elasticity effect on m is also ruled out since for viscoelastic droplet transient deformation (similar to our study), negligible effects of elasticity on deformation have been reported in the literature⁴⁰. Thus, Eq. 1 can be extended, in principle, to other liquid crystalline tactoids based on mesogens with different elasticity from that of amyloid fibrils.”

4. After Eq. (2) can you also explain what θ is in that equation? Further down (line 246) you mention the values that it can take—can you also explain why?

In the revised manuscript, we now both explained what θ is and why it assumes the values mentioned in the text.

To explain what θ is, we added on page 12:

“The symbol θ represents the twist term equal to $n \cdot \nabla \times n$ in the Frank–Oseen elasticity theory with n the nematic director.”

And, to explain why it takes the mentioned values in the text. we added on page 12:

“Note that θ assumes the value of the $-q_\infty$ for the cholesteric tactoids as the free energy of the cholesteric tactoids is in the lowest state when $\theta = -q_\infty$; and θ is zero for homogenous tactoids as there is no twist for the director field, i.e. $n \cdot \nabla \times n = 0$.”

5. Some Editorial Remarks

- a) In line 5, use “follows” instead of “follow”**
- b) In line 218, use “remains” instead of “remain”**
- c) In line 219, use “is tested” instead of “are tested”**
- d) In figure 4, can you explain what α_0 is? (equilibrium aspect ratio?)**
- e) In line 348, can you also add the symbol for the entrance length?**
- f) The year of publication for reference 1 should be 2019 not 2018.**

We would like to thank the Reviewer for pointing out these errors. In the revised manuscript, we fixed all above points. In detail:

- a. We used “follows”. (However, we find this error in line 164)
- b. We used “remains”.
- c. We used “is tested”.
- d. In Figure 4 caption, we added that “The term α_0 denotes the initial aspect ratio of the tactoids, see SI for argument allowing the estimation of α_0 based on the equilibrium aspect ratio of the tactoids.”
- e. We add the symbol for the entrance length in line 348 as “...while different entrance lengths (l_e) were used...”
- f. We corrected the year in reference 1 to 2019.

REVIEWERS' COMMENTS

Reviewer #2 (Remarks to the Author):

I have carefully examined the provided author response letter and the new version of the manuscript & related materials. The authors have taken into account all my comments and improved the manuscript and the corresponding supplementary material. The manuscript is now ready for publication in the Nature Communications.

Reviewer #3 (Remarks to the Author):

As I have found that the authors have satisfactorily responded to all my previous comments and they have made appropriate changes to their manuscript, I have no further objections to its publication.

Reviewer #2

I have carefully examined the provided author response letter and the new version of the manuscript & related materials. The authors have taken into account all my comments and improved the manuscript and the corresponding supplementary material. The manuscript is now ready for publication in the Nature Communications.

We would like to thank the Reviewer #2 for his/her insightful review and constructive comments that helped to improve our manuscript.

Reviewer #3

As I have found that the authors have satisfactorily responded to all my previous comments and they have made appropriate changes to their manuscript, I have no further objections to its publication.

We would like to thank the Reviewer #3 for his/her insightful review and constructive comments that helped to improve our manuscript.